# Hierarchical progressive learning of cell identities in single-cell data

Lieke Michielsen [1,2,3], Marcel J. T. Reinders [1,2,3] & Ahmed Mahfouz [1,2,3]✉

Supervised methods are increasingly used to identify cell populations in single-cell data. Yet, current methods are limited in their ability to learn from multiple datasets simultaneously, are hampered by the annotation of datasets at different resolutions, and do not preserve annotations when retrained on new datasets. The latter point is especially important as researchers cannot rely on downstream analysis performed using earlier versions of the dataset. Here, we present scHPL, a hierarchical progressive learning method which allows continuous learning from single-cell data by leveraging the different resolutions of annotations across multiple datasets to learn and continuously update a classification tree. We evaluate the classification and tree learning performance using simulated as well as real datasets and show that scHPL can successfully learn known cellular hierarchies from multiple datasets while preserving the original annotations. scHPL is available at https://github.com/lcmmichielsen/scHPL.

¹ Department of Human Genetics, Leiden University Medical Center, Leiden, The Netherlands. ² Leiden Computational Biology Center, Leiden University Medical Center, Leiden, The Netherlands. ³ Delft Bioinformatics Lab, Delft University of Technology, Delft, The Netherlands. ✉email: a.mahfouz@lumc.nl

Cell identification is an essential step in single-cell studies with profound effects on downstream analysis. For example, in order to compare cell-population-specific eQTL findings across studies, cell identities should be consistent[1]. Currently, cells in single-cell RNA-sequencing (scRNA-seq) datasets are primarily annotated using clustering and visual exploration techniques, i.e., cells are first clustered into populations that are subsequently named based on the expression of marker genes. This is not only time-consuming but also subjective[2]. The number of cell populations identified in a dataset, for example, is strongly correlated with the number of cells analyzed, which results in inconsistency across datasets[3–5].

Recently, many supervised methods have been developed to replace unsupervised techniques. The underlying principles of these methods vary greatly. Some methods, for instance, rely on prior knowledge and assume that for each cell population marker genes can be defined (e.g., SCINA[6] and Garnett[7]), while others search for similar cells in a reference database (e.g., scmap[8] and Cell-BLAST[9]), or train a classifier using a reference atlas or a labeled dataset (e.g., scPred[10] and CHETAH[11]).

Supervised methods rely either on a reference atlas or labeled dataset. Ideally, we would use a reference atlas containing all possible cell populations to train a classifier. Such an atlas, however, does not exist yet and might never be fully complete. In particular, aberrant cell populations might be missing as a huge number of diseases exist and mutations could result in new cell populations. To overcome these limitations, some methods (e.g., OnClass) rely on the Cell Ontology to identify cell populations that are missing from the training data but do exist in the Cell Ontology database[12]. These Cell Ontologies, however, were not developed for scRNA-seq data specifically. As a consequence, many newly identified (sub)populations are missing and relationships between cell populations might be inaccurate. A striking example of this inadequacy is neuronal cell populations. Recent single-cell studies have identified hundreds of populations[4,13,14], including seven subtypes and 92 cell populations in one study only[5]. In contrast, the Cell Ontology currently includes only one glutamatergic neuronal cell population without any subtypes.

Since no complete reference atlas is available, a classifier should ideally be able to combine the information of multiple annotated datasets and continue learning. Each time a new cell population is found in a dataset, it should be added to the knowledge of the classifier. We advocate that this can be realized with progressive learning, a learning strategy inspired by humans. Human learning is a continuous process that never ends[15]. Using progressive learning, the task complexity is gradually increased, for instance, by adding more classes, but it is essential that the knowledge of the previous classes is preserved[16,17]. This strategy allows combining information of multiple existing datasets and retaining the possibility to add more datasets afterward. However, it cannot be simply applied to scRNA-seq datasets as a constant terminology to describe cell populations is missing, which eliminates straightforward identification of new cell populations based on their names. For example, the recently discovered neuronal populations are typically identified using clustering and named based on the expression of marker genes. A standardized nomenclature for these clusters is missing[18], so the relationship between cell populations defined in different datasets is often unknown.

Moreover, the level of detail (resolution) at which datasets are annotated highly depends on the number of cells analyzed[19]. For instance, if a dataset is annotated at a low resolution, it might contain T cells, while a dataset at a higher resolution can include subpopulations of T cells, such as CD4+ and CD8+ T cells. We need to consider this hierarchy of cell populations in our representation, which can be done with a hierarchical classifier. This has the advantage that cell population definitions of multiple datasets can be combined, ensuring consistency. A hierarchical classifier has additional advantages in comparison to a classifier that does not exploit this hierarchy between classes (here denoted as "flat classifier"). One of these advantages is that the classification problem is divided into smaller sub-problems, while a flat classifier needs to distinguish between many classes simultaneously. Another advantage is that if we are not sure about the annotation of an unlabeled cell at the highest resolution, we can always label it as an intermediate cell population (i.e., at a lower resolution).

Currently, some classifiers, such as Garnett, CHETAH, and Moana, already exploit this hierarchy between classes[7,11,20]. Garnett and Moana both depend on prior knowledge in the form of marker genes for the different classes. Especially for deeper annotated datasets, it can be difficult to define marker genes for each cell population that are robust across scRNA-seq datasets[21,22]. Moreover, we have previously shown that adding prior knowledge is not beneficial[23]. CHETAH, on the contrary, constructs a classification tree based on one dataset by hierarchically clustering the reference profiles of the cell populations and classifies new cells based on the similarity to the reference profile of that cell population. However, simple flat classifiers outperform CHETAH[23], indicating that a successful strategy to exploit this hierarchy is still missing. Furthermore, these hierarchical classifiers cannot exploit the different resolutions of multiple datasets as this requires adaptation of the hierarchical representation.

Even if multiple datasets are combined into a hierarchy, there might be unseen populations in an unlabeled dataset. Identifying these cells as a new population is a challenging problem. Although some classifiers have implemented an option to reject cells, they usually fail when being tested in a realistic scenario[23]. In most cases, the rejection option is implemented by setting a threshold on the posterior probability[7,10,23,24]. If the highest posterior probability does not exceed a threshold, the cell is rejected. By looking at the posterior, the actual similarity between a cell and the cell population is ignored.

In this work, we propose a hierarchical progressive learning approach to overcome these challenges. To summarize our contributions: (i) we exploit the hierarchical relationships between cell populations to be able to classify data sets at different resolutions, (ii) we propose a progressive learning approach that updates the hierarchical relationships dynamically and consistently, and (iii) we adopt an advanced rejection procedure including a one-class classifier to be able to discover new cell (sub)populations.

## Results

**Hierarchical progressive learning of cell identities.** We developed scHPL, a hierarchical progressive learning approach to learn a classification tree using multiple labeled datasets (Fig. 1A) and use this tree to predict the labels of a new, unlabeled dataset (Fig. 1B). The tree is learned using multiple iterations (Methods). First, we match the labels of two datasets by training a flat classifier for each dataset and predicting the labels of the other dataset. Based on these predictions we create a matching matrix ($X$) and match the cell populations of the two datasets. In the matching process, we separate different biological scenarios, such as a perfect match, merging or splitting cell populations, as well as creating a new population (Fig. 2, Supplementary Table 1). In the following iterations, we add one labeled dataset at a time by training a flat classifier on this new dataset and training the previously learned tree on all pre-existing datasets. Similar to the previous iteration, the tree is updated after cross-prediction and matching of the labels. It could happen that datasets are

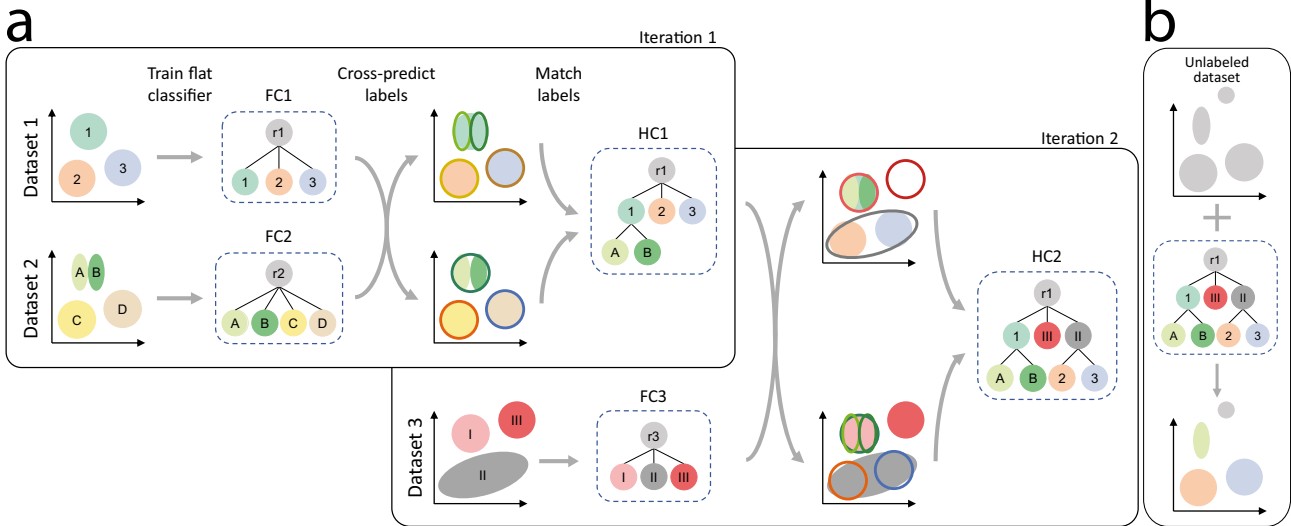

**Fig. 1 Schematic overview of scHPL. a** Overview of the training phase. In the first iteration, we start with two labeled datasets. The colored areas represent the different cell populations. For both datasets a flat classifier (FC1 and FC2) is constructed. Using this tree and the corresponding dataset, a classifier is trained for each node in the tree except for the root. We use the trained classification tree of one dataset to predict the labels of the other. The decision boundaries of the classifiers are indicated with the contour lines. We compare the predicted labels to the cluster labels to find matches between the labels of the two datasets. The tree belonging to the first dataset is updated according to these matches, which results in a hierarchical classifier (HC1). In dataset 2, for example, subpopulations of population "1" of dataset 1 are found. Therefore, these cell populations, "A" and "B", are added as children to the "1" population. In iteration 2, a new labeled dataset is added. Again a flat classifier (FC3) is trained for this dataset and HC1 is trained on datasets 1 and 2, combined. After cross-prediction and matching the labels, we update the tree which is then trained on all datasets 1–3 (HC2). **b** The final classifier can be used to annotate a new unlabeled dataset. If this dataset contains unknown cell populations, these will be rejected.

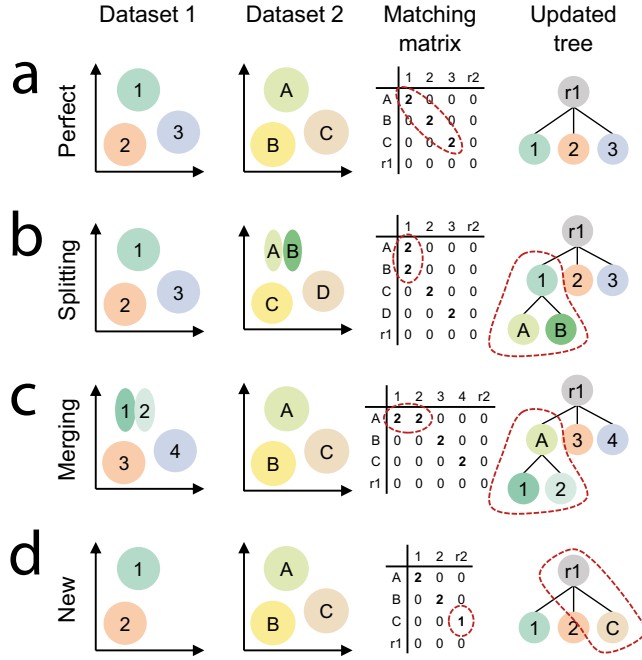

**Fig. 2 Schematic examples of different matching scenarios. a** Perfect match, **b** splitting, **c** merging, and **d** new population. The first two columns represent a schematic representation of two datasets. After cross-predictions, the matching matrix ($X$) is constructed using the confusion matrices (Methods). We update the tree based on $X$.

inconsistently labeled and the labels cannot be matched (Supplementary Note 1). In this case, one of the populations might be missing from the tree.

Either during tree learning or prediction, there can be unseen populations. Therefore, an efficient rejection option is needed,

which we do in two steps. First, we reject cells by thresholding the reconstruction error of a cell when applying a PCA-based dimension reduction: a new, unknown, population is not used to learn the PCA transformation, and consequently will not be properly represented by the selected PCs, leading to a high reconstruction error (Methods). Second, to accommodate rejections when the new population is within the selected PCA domain, scHPL adopts two alternatives to classify cells: a linear and a one-class support vector machine (SVM). The linear SVM has shown high performance in a benchmark of scRNA-seq classifiers[23] but has a limited rejection option. Whereas, the one-class SVM solves this as only positive training samples are used to fit a tight decision boundary[25].

**Linear SVM has a higher classification accuracy than one-class SVM.** We tested our hierarchical classification scheme by measuring the classification performance of the one-class SVM and linear SVM on simulated, PBMC (PBMC-FACS) and brain (Allen Mouse Brain (AMB)) data using 10-, 10-, and 5-fold cross-validation respectively (Methods). The simulated dataset was constructed using Splatter[26] and consists of 8839 cells, 9000 genes, and 6 different cell populations (Supplementary Fig. 1). PBMC-FACS is the downsampled FACS-sorted PBMC dataset from Zheng et al.[27] and consists of 20,000 cells and 10 cell populations. The AMB dataset is challenging as it has deep annotation levels[5], containing 92 different cell populations ranging in size from 11 to 1348 cells. In these experiments, the classifiers were trained on predefined trees (Supplementary Figs. 1–3).

On all datasets, the linear SVM performs better than the one-class SVM (Fig. 3A–D). The simulated dataset is relatively easy since the cell populations are widely separated (Supplementary Fig. 1C). The linear SVM shows an almost perfect performance: only 0.9% of the cells are rejected (based on the reconstruction error only), which is in line with the adopted threshold resulting in 1% false negatives. The one-class SVM labels 92.9% of the cells

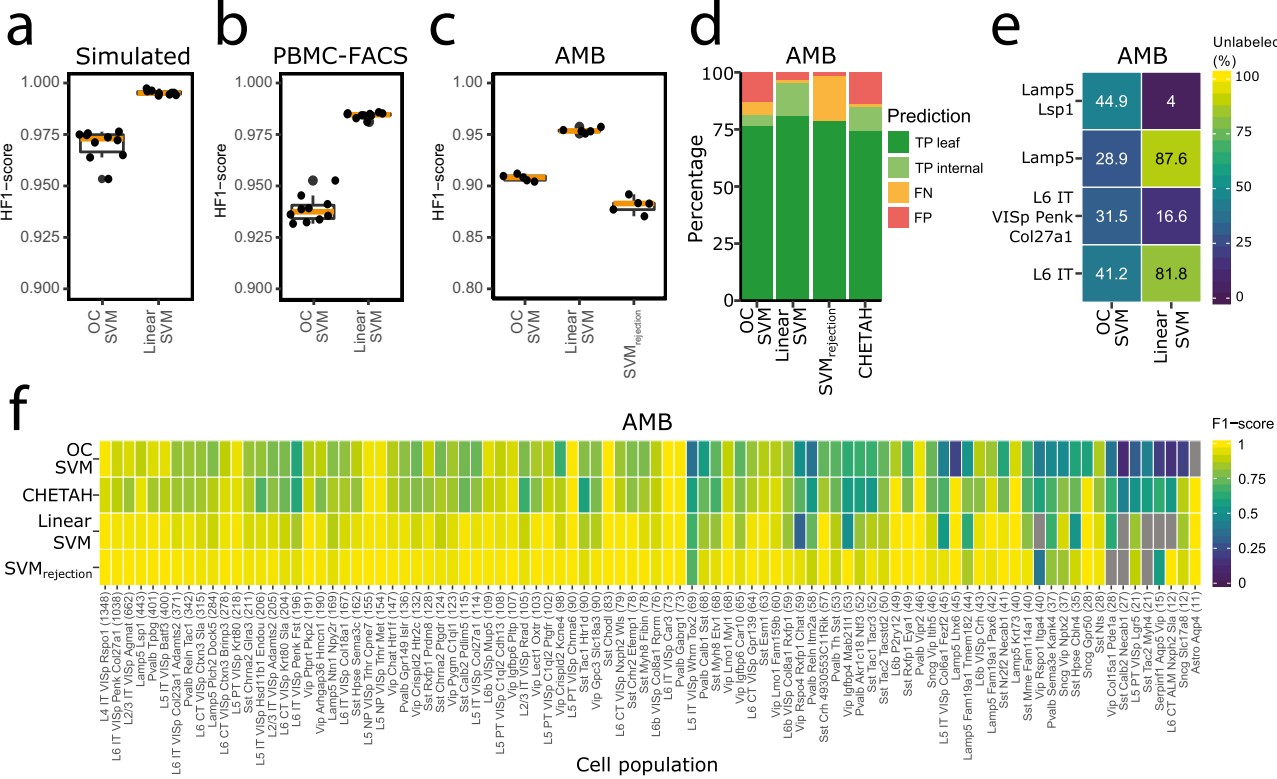

**Fig. 3 Classification performance. a–c** Boxplots showing the HF1-score of the one-class and linear SVM during *n*-fold cross-validation on the **a** simulated (*n* = 10), **b** PBMC-FACS (*n* = 10), and **c** AMB (*n* = 5) dataset. In the boxplots, the middle (orange) line represents the median, the lower and upper hinge represents the first and third quartiles, and the lower and upper whiskers represent the values no further than 1.5 interquartile range away from the lower and upper hinge, respectively. **d** Barplot showing the percentage of true positives (TP), false negatives (FN), and false positives (FP) per classifier on the AMB dataset. For the TPs a distinction is made between correctly predicted leaf nodes and internal nodes. **e** Heatmap showing the percentage of unlabeled cells per classifier during the different rejection experiments. **f** Heatmap showing the F1-score per classifier per cell population on the AMB dataset. Gray indicates that a classifier never predicted a cell to be of that population.

correctly, the rest is labeled as an internal node (2.3%) or rejected (4.8%), which results in a median Hierarchical F1-score (HF1-score) of 0.973, where HF1 is an F1-score that considers class importance across the hierarchy (Methods).

As expected, the performance of the classifiers on real data drops, but the HF1-scores remain higher than 0.9. On both the PBMC-FACS and AMB dataset, the performance of the linear SVM is higher than the one-class SVM (Fig. 3B–D). For the AMB dataset, we used the same cross-validation folds as in Abdelaal et al.[23], which enables us to compare our results. When comparing to CHETAH, which allows hierarchical classification, we notice that the linear SVM performs better based on the median F1-score (0.94 vs. 0.83). The one-class SVM has a slightly lower median F1-score (0.80 vs. 0.83), but it has more correctly predicted cells and less wrongly predicted cells (Fig. 3D).

The linear (hierarchical) SVM also shows a better performance compared to SVM_rejection, which is a flat linear SVM with a rejection option based on the posterior probability and was the best classifier for this data[23]. SVM_rejection, however, has a slightly higher median F1-score (0.98 vs. 0.94). This is mainly because it makes almost no mistakes, only 1.7% of the cells are wrongly labeled (Fig. 3D). The number of rejected cells, on the other hand, is not considered when calculating the median F1-score. This number is relatively high for SVM_rejection (19.8%). The linear SVM, on the contrary, has almost no rejected cells, which is also reflected in a higher HF1-score (Fig. 3C). Because of this large amount of rejections of SVM_rejection, the one-class SVM also has a higher HF1-score.

On the AMB dataset, we observe that the performance of all classifiers decreases when the number of cells per cell population

becomes smaller. The performance of the one-class SVM is affected more than the others (Fig. 3F). The one-class SVM, for instance, never predicts the "Astro Aqp4" cells correctly, while this population is relatively different from the rest as it is the only non-neuronal population. This cell population, however, only consists of eleven cells.

In the previous experiments, we used all genes to train the classifiers. Since the selection of highly variable genes (HVGs) is common in scRNA-seq analysis pipelines, we tested the effect of selecting HVGs on the classification performance of the PBMC-FACS dataset. We noted that using all genes results in the highest HF1-score for both the linear and one-class SVM (Supplementary Fig. 4).

**One-class SVM detects new cells better than linear SVM.** Besides a high accuracy, the classifiers should be able to reject unseen cell populations. First, we evaluated the rejection option on the simulated data. In this dataset, the cell populations are distinct, so we expect that this is a relatively easy task. We removed one cell population, "Group 3", from the training set and used this population as a test set. The one-class SVM outperforms the linear SVM as it correctly rejects all these cells, while the linear SVM rejects only 38.9% of them.

Next, we tested the rejection option on the AMB dataset. Here, we did four experiments and each time removed a node, including all its subpopulations, from the predefined tree (Supplementary Fig. 3). We removed the "L6 IT" and "Lamp5" cell populations from the second layer of the tree, and the "L6 IT

VISp Penk Col27a1" and "Lamp5 Lsp1" from the third layer. When a node is removed from the second layer of the tree, the linear SVM clearly rejects these cells better than the one-class SVM (Fig. 3E). On the contrary, the one-class SVM rejects leaf node cells better.

**scHPL accurately learns cellular hierarchies**. Next, we tested if we could learn the classification trees for the simulated and PBMC-FACS data using scHPL. In both experiments, we performed 10-fold cross-validation and split the training set into three different batches, Batches 1–3, to simulate the idea of different datasets. To obtain different annotation levels in these batches, multiple cell populations were merged into one population in some batches, or cell populations were removed from certain batches (Tables S2 and S3). Batch 1 contains the lowest resolution and Batch 3 the highest. When learning the trees, we try all (six) different orders of the batches to see whether this affects tree learning. Combining this with the 10-fold cross-validation, 60 trees were learned in total by each classifier. To summarize the results, we constructed a final tree in which the thickness of an edge indicates how often it appeared in the 60 learned trees.

The linear and one-class SVM showed stable results during both experiments; all 60 trees—except for two trees learned by the one-class SVM on the PBMC data—look identical (Fig. 4A–D). The final tree for the simulated data looks as expected, but the tree for the PBMC data looks slightly different from the predefined hematopoietic tree (Supplementary Fig. 2A). In the learned trees, CD4+ memory T cells are a subpopulation of CD8+ instead of CD4+ T cells. The correlation between the centroids of CD4+ memory T cells and CD8+ T cells ($r = 0.985 \pm 0.003$) is also slightly higher than the correlation to CD4+ T cells ($r = 0.975 \pm 0.002$) (Supplementary Fig. 5). Using the learned tree instead of the predefined hematopoietic tree improves the classification performance of the linear SVM slightly (HF1-score = 0.990 vs. 0.985). Moreover, when relying on the predefined hematopoietic tree, CD4+ memory T cells, CD8+ T cells, and CD8+ naive T cells were also often confused, further highlighting the difficulty in distinguishing these populations based on their transcriptomic profiles alone (Tables S4 and 5).

Next, we tested the effect of the matching threshold (default = 0.25) on the tree construction by varying this to 0.1 and 0.5. For the linear SVM, changing the threshold had no effect. For the one-class SVM, we noticed a small difference when changing the threshold to 0.1. The two trees that were different using the default threshold (Fig. 4D), were now constructed as the other 58 trees. In general, scHPL is robust to settings of the matching threshold due to its reliance on reciprocal classification.

**Missing populations affect tree construction with linear SVM**. We tested whether new or missing cell populations in the training set could influence tree learning. We mimicked this scenario using the simulated dataset and the same batches as in the previous tree learning experiment. In the previous experiment, "Group5" and "Group6" were merged into "Group56" in Batch 2, but now we removed "Group5" completely from this batch (Supplementary Table 6). In this setup, the linear SVM misconstructs all trees (Fig. 4E). If "Group5" is present in one batch and absent in another, the "Group5" cells are not rejected, but labeled as "Group6". Consequently, "Group6" is added as a parent node to "Group5" and "Group6". On the other hand, the one-class SVM suffers less than the linear SVM from these missing populations and correctly learns the expected tree in two-

third of the cases (Fig. 4F). In the remaining third (20 trees), "Group5" matched perfectly with "Group456" and was thus not added to the tree. This occurs only if we have the following order: Batch 1–Batch 3–Batch 2 or Batch 3–Batch 1–Batch 2. Adding batches in increasing or decreasing resolution consequently resulted in the correct tree.

**Linear SVM can learn the classification tree during an inter-dataset experiment**. Finally, we tested scHPL in a realistic scenario by using three PBMC datasets (PBMC-eQTL, PBMC-Bench10Xv2, and PBMC-FACS) to learn a classification tree and using this tree to predict the labels of a fourth PBMC dataset (PBMC-Bench10Xv3) (Table 1). Before applying scHPL, we aligned the datasets using Seurat[28]. We constructed an expected classification tree based on the names of the cell populations in the datasets (Fig. 5A). Note that matching based on names might result in an erroneous tree since every dataset was labeled using different clustering techniques, marker genes, and their own naming conventions.

When comparing the tree learned using the linear SVM to the expected tree, we notice five differences (Fig. 5A, B). Some of these differences are minor, such as the matching of monocytes from the Bench10Xv2 dataset to myeloid dendritic cells (mDC), CD14+ monocytes, and the CD16+ monocytes. Monocytes can differentiate into mDC which can explain their transcriptomic similarity[29]. Other differences between the reconstructed and the expected tree are likely the result of (partly) mislabeled cell populations in the original datasets (Supplementary Figs. 6–15). (i) According to the expression of *FCER1A* (a marker for mDC) and *FCGR3A* (CD16+ monocytes), the labels of the mDC and the CD16+ monocytes in the eQTL dataset are reversed (Supplementary Figs. 6–8). (ii) Part of the CD14+ monocytes in the FACS dataset express *FCER1A*, which is a marker for mDC (Supplementary Figs. 6, 8, and 9). The CD14+ monocytes in the FACS dataset are thus partly mDCs, which explains the match with the mDC from the eQTL dataset. (iii) Part of the CD4+ T cells from the eQTL and Bench10Xv2 dataset should be relabeled as CD8+ T cells (Supplementary Figs. 6, and 10–13). In these datasets, the cells labeled as the CD8+ T cells only contain cytotoxic CD8+ T cells, while naive CD8+ T cells are all labeled as CD4+ T cells. This mislabeling explains why the CD8 + naive T cells are a subpopulation of the CD4+ T cells. (iv) Part of the CD34+ cells in the FACS dataset should be relabeled as pDC (Supplementary Figs. 6, 14, and 15), which explains why the pDC are a subpopulation of the CD34+ cells. In the FACS dataset, the labels were obtained using sorting, which would indicate that these labels are correct. The purity of the CD34+ cells, however, was significantly low (45%) compared to other cell populations (92–100%)[27]. There is only one difference, however, that cannot be explained by mislabeling. The NK cells from the FACS dataset do not only match the NK cells from the eQTL dataset, but also the CD8+ T cells.

Most cells of the Bench10Xv3 dataset can be correctly annotated using the learned classification tree (Fig. 5E). Interestingly, we notice that the CD16+ monocytes are predicted to be mDCs and vice versa, which could be explained by the fact that the labels of the mDCs and the CD16+ monocytes were flipped in the eQTL dataset. Furthermore, part of the CD4+ T cells are predicted to be CD8+ naïve T cells. In the Bench10Xv3, we noticed the same mislabeling of part of the CD4+ T cells as in the eQTL and Bench10Xv2 datasets, which supports our predictions (Supplementary Figs. 6 and 10–13).

The tree constructed using the one-class SVM differs slightly compared to the linear SVM (Supplementary Fig. 16A). Here, the monocytes from the Bench10Xv2 match the CD14+ monocytes

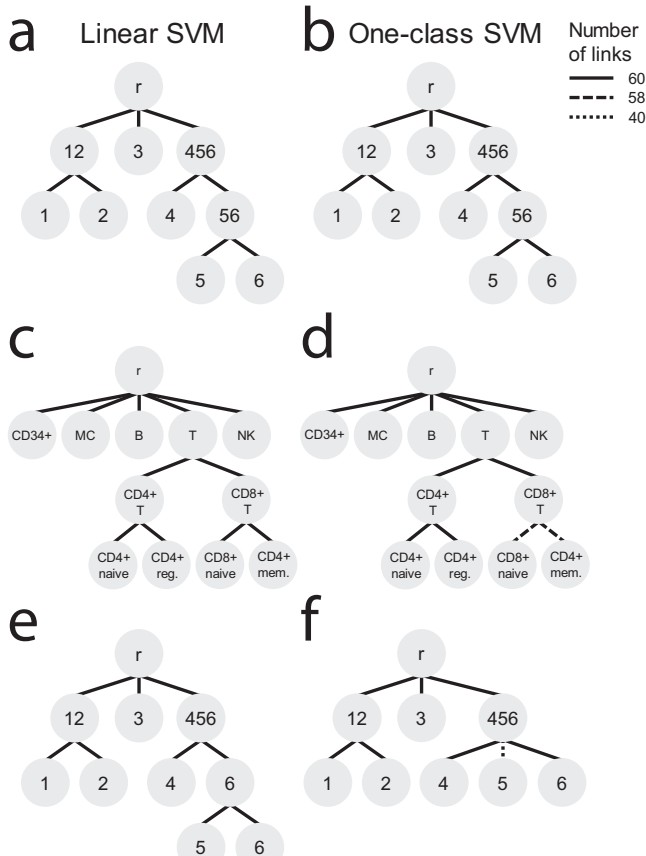

**Fig. 4 Tree learning evaluation.** Classification trees learned when using a **a**, **c**, **e** linear SVM or **b**, **d**, **f** one-class SVM during the **a**, **b** simulated, **c**, **d** PBMC-FACS, and **e**, **f** simulated rejection experiment. The line pattern of the links indicates how often that link was learned during the 60 training runs. **d** In 2/60 trees, the link between the CD8+ T cells and the CD8+ naive and CD4+ memory T cells is missing. In those trees, the CD8+ T cells and CD8+ naive T cells have a perfect match and the CD4+ memory T cells are missing from the tree. **f** In 20/60 trees, the link between "Group456" and "Group5" is missing. In those trees, these two populations are a perfect match.

and mDC (which are actually CD16+ monocytes) as we would expect. Next, the CD14+ monocytes from the FACS dataset merge the CD16+ monocytes (which are actually mDC) and the monocytes. Using the one-class SVM the CD8+ T cells and NK cells from the Bench10Xv2 dataset are missing since there was a complex scenario. The NK cells are a relatively small population in this dataset which made it difficult to train a classifier for this population.

In the previous experiments, we used the default setting of Seurat to align the datasets (using 2000 genes). We tested whether changing the number of genes to 1000 and 5000 affects the performance. When using the one-class SVM, the number of genes does not affect tree construction. For the linear SVM, we notice one small difference when using 1000 genes: the CD8+ T cells from the Bench10Xv2 dataset are a subpopulation of the CD8+ T cells from the eQTL dataset instead of a perfect match.

The predicted labels of the Bench10Xv3 dataset change slightly when using a different number of genes (Supplementary Fig. 17). Whether more genes improve the prediction, differs per cell population. The labels of the megakaryocytes, for instance, are better predicted when more genes are used, but for the dendritic cells we observe the reverse pattern.

**Mapping brain cell populations using scHPL**. Next, we applied scHPL to construct a tree that maps the relationships between brain cell populations. This is a considerably more challenging task compared to PBMCs given the large number of cell populations as well as the fact that brain cell types are not consistently annotated. First, we combined two datasets from the primary visual cortex of the mouse brain, AMB2016 and AMB2018[4,5]. AMB2018 contains more cells (12,771 vs. 1298) and is clustered at a higher resolution (92 cell populations vs. 41) compared to AMB2016. Before applying scHPL, we aligned the datasets using Seurat[28]. Using scHPL with the linear SVM results in an almost perfect tree (Fig. 6). We verified these results by comparing our constructed tree to cluster correspondences in Extended Data Fig. 6 from Tasic et al.[5]. Since AMB2018 is clustered at a higher resolution, most populations are subpopulations of AMB2016, which are all correctly identified by scHPL. Conversely, three L4 populations from AMB2016 were merged into one population (L4 IT VISp Rspo1) from AMB2018[5], forming a continuous spectrum. This relation was also automatically identified using scHPL (Fig. 6). Compared to the results from Tasic et al.[5], one cell population from AMB2018 is attached to a different parent node. scHPL assigned "L6b VISp Col8a1 Rprm" as a subpopulation of "L6a Sla" instead of "L6b Rgs12". This population, however, does not express *Rgs12*, but does express *Sla* (Supplementary Fig. 18), supporting the matching identified by scHPL. Three cell populations could not be added to the tree due to complex scenarios. According to Extended Data Fig. 6 from Tasic et al.[5], these AMB2018 populations are a subpopulation of multiple AMB2016 subpopulations.

The AMB2016 and AMB2018 datasets were generated and analyzed by the same group and hence the cluster matching is certainly easier than a real-life scenario. Therefore, we tested scHPL also on a complicated scenario with brain datasets that are sequenced using different protocols and by different labs (Supplementary Table 7, Supplementary Fig. 19). We used three datasets (Zeisel, Tabula Muris, and Saunders) to construct the tree (Fig. 7A–D)[2,30,31]. Before applying scHPL, we aligned the datasets using Seurat[28]. The Zeisel dataset is annotated at two resolutions. First, we constructed a tree using a linear SVM based on the low resolution of Zeisel. We started with the Saunders dataset and added Zeisel (Fig. 7E). This is a clear illustration of the possible scenarios scHPL can manage. Some populations are a perfect match between the two datasets (e.g., neurons), some populations from Saunders are split (e.g., astrocytes), some are merged (e.g., macrophages and microglia), and some populations from Zeisel have no match (e.g., Ttr). Next, we updated the tree by adding the Tabula Muris dataset (Fig. 7F). Here, we found matches that would not have been possible to identify by relying on the assigned cell type labels to map cell types. For example, mural cells from Saunders are a perfect match with the brain pericytes from the Tabula Muris. The results of scHPL with the one-class SVM were almost identical to the linear SVM (Supplementary Fig. 20A).

Next, we used the resulting tree to predict the labels of a fourth independent dataset (Rosenberg)[32]. The predictions from the linear and the one-class SVM are very similar (Figs. 7G and S20B). The only difference is that the linear SVM correctly predicts some progenitor or precursor neuronal populations from Rosenberg to be "neurogenesis" while the one-class SVM rejects these populations.

To assess the effect of the annotation resolution, we repeated the analysis using the higher resolution annotation from the Zeisel dataset (Supplementary Figs. 21–23). Here, we noticed that the "brain pericytes (TM)" and "pericytes (Zeisel)"—two populations one would easily match based on the names only—are not in the same subtree. "Brain pericyte (TM)" forms a perfect match

**Table 1 Number of cells per cell population in the different training datasets (batches) and test dataset. Subpopulations are indicated using an indent.**

| Cell population | Batch 1 eQTL | Batch 2 Bench-10Xv2 | Batch 3 FACS | Test dataset Bench-10Xv3 |
|---|---|---|---|---|
| CD19+ B | 812 | 676 | 2000 | 346 |
| CD34+ | | | 2000 | |
| Monocytes (MC) | | 1194 | | |
| CD14+ | 2081 | | 2000 | 354 |
| CD16+ | 274 | | | 98 |
| CD4+ T | 13,523 | 1458 | | 960 |
| Reg. | | | 2000 | |
| Naive | | | 2000 | |
| Memory | | | 2000 | |
| CD8+ T | 4195 | 2128 | | 962 |
| Naive | | | 2000 | |
| Megakaryocyte (MK) | 142 | 433 | | 270 |
| NK cell | | 429 | 2000 | 194 |
| CD56+ bright | 355 | | | |
| CD56+ dim | 2415 | | | |
| Dendritic | | | | 35 |
| | 101 | | | |
| Plasmacytoid (pDC) | | | | |
| Myeloid (mDC) | 455 | | | |

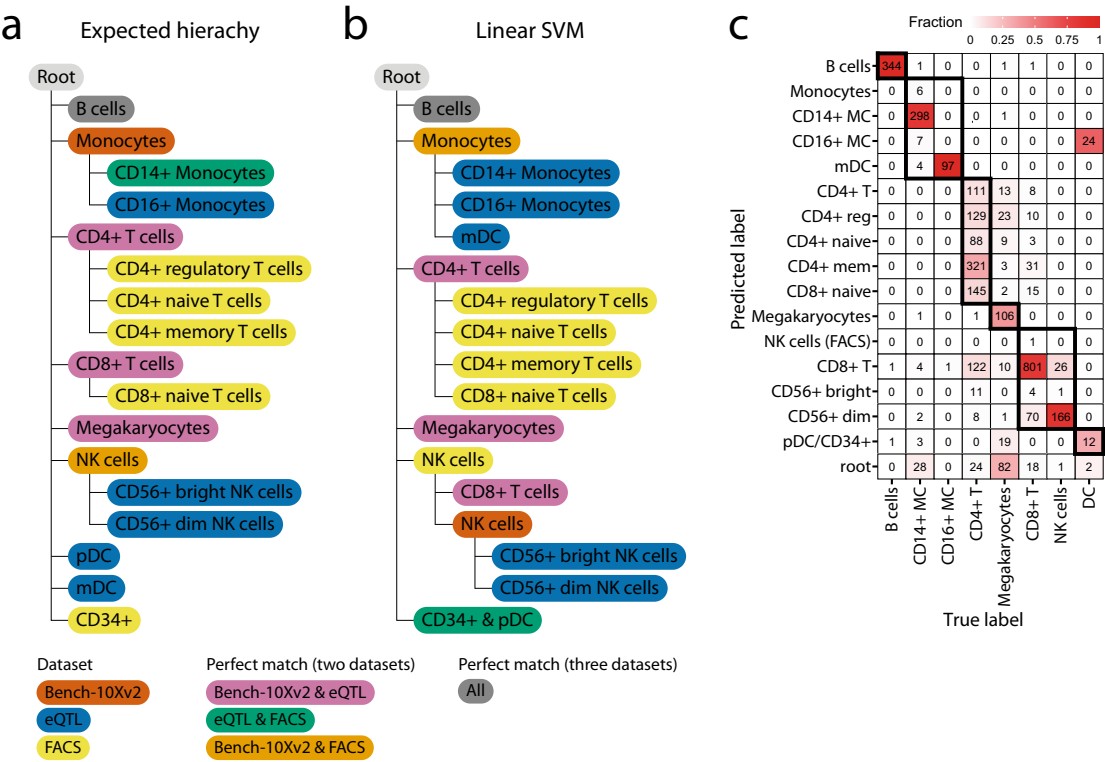

**Fig. 5 PBMC inter-dataset evaluation. a** Expected and **b** learned classification tree when using a linear SVM on the PBMC datasets. The color of a node represents the agreement between dataset(s) regarding that cell population. **c** Confusion matrix when using the learned classification tree to predict the labels of PBMC-Bench10Xv3. The dark boundaries indicate the hierarchy of the constructed classification tree.

with "mural (Saunders)" and "vascular smooth muscle cells (Zeisel)", while "pericytes (Zeisel)" is a subpopulation of "endothelial stalk (Saunders)" and "endothelial cell (TM)" (Supplementary Figs. 22 and 23). In the UMAP embedding of the integrated datasets, the "pericytes" and "brain pericyte" are at different locations, but they do overlap with the cell populations they were matched with (Supplementary Fig. 21). This highlights the power of scHPL matching rather than name-based matching.

## Discussion

In this study, we showed that scHPL can learn cell identities progressively from multiple reference datasets. We showed that using our approach the labels of two AMB datasets can successfully be matched to create a hierarchy containing mainly neuronal cell populations and that we can combine three other brain datasets to create a hierarchy containing mainly nonneuronal cell populations. In both experiments, we discovered

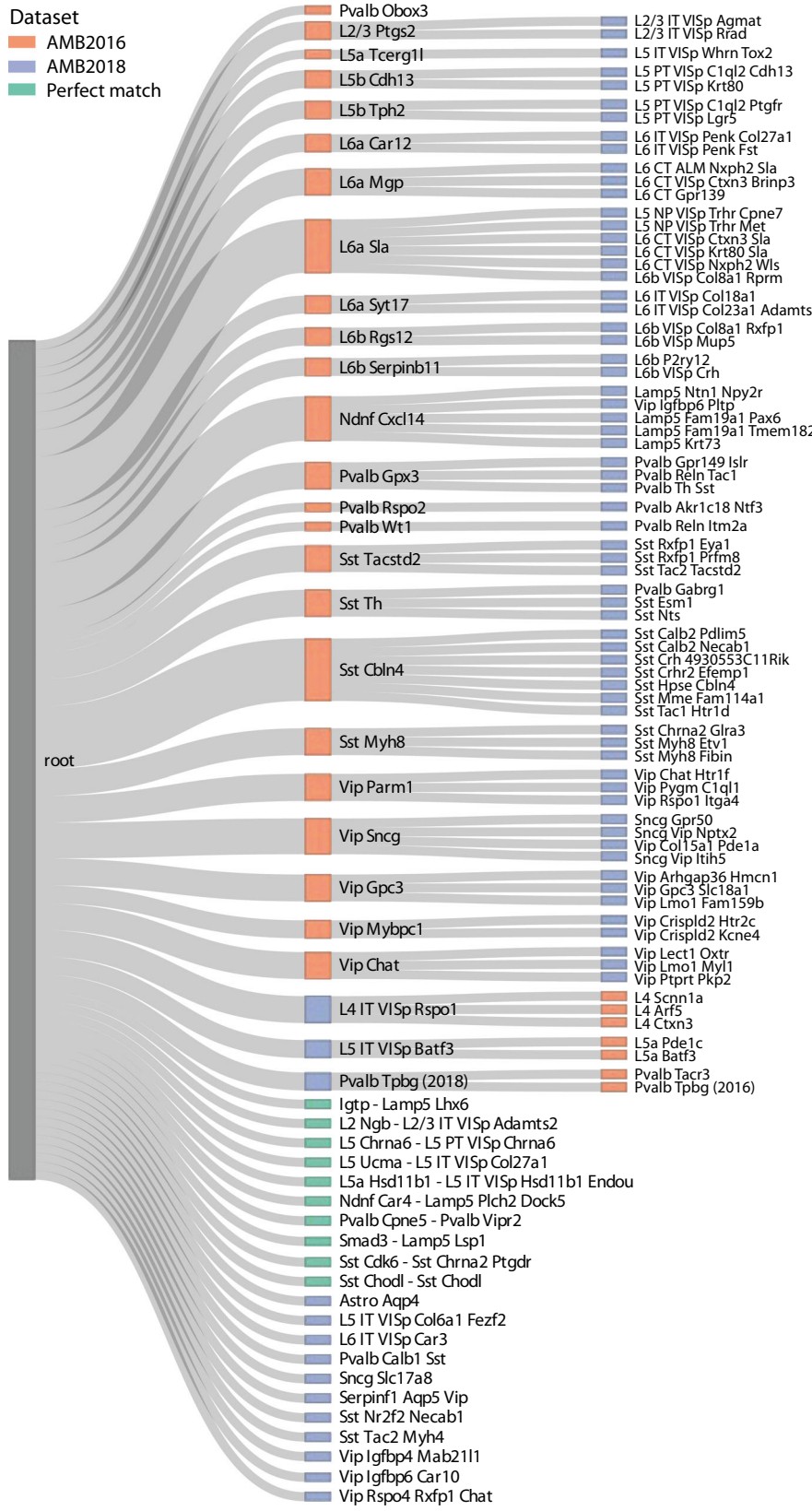

**Fig. 6 Constructed hierarchy for the AMB datasets.** Learned classification tree after applying scHPL with a linear SVM on the AMB2016 and AMB2018 datasets. A green node indicates that a population from the AMB2016 and AMB2018 datasets had a perfect match. Three populations from the AMB2018 dataset are missing from the tree: "Pvalb Sema3e Kank4", "Sst Hpse Sema3c", and "Sst Tac1 Tacr3".

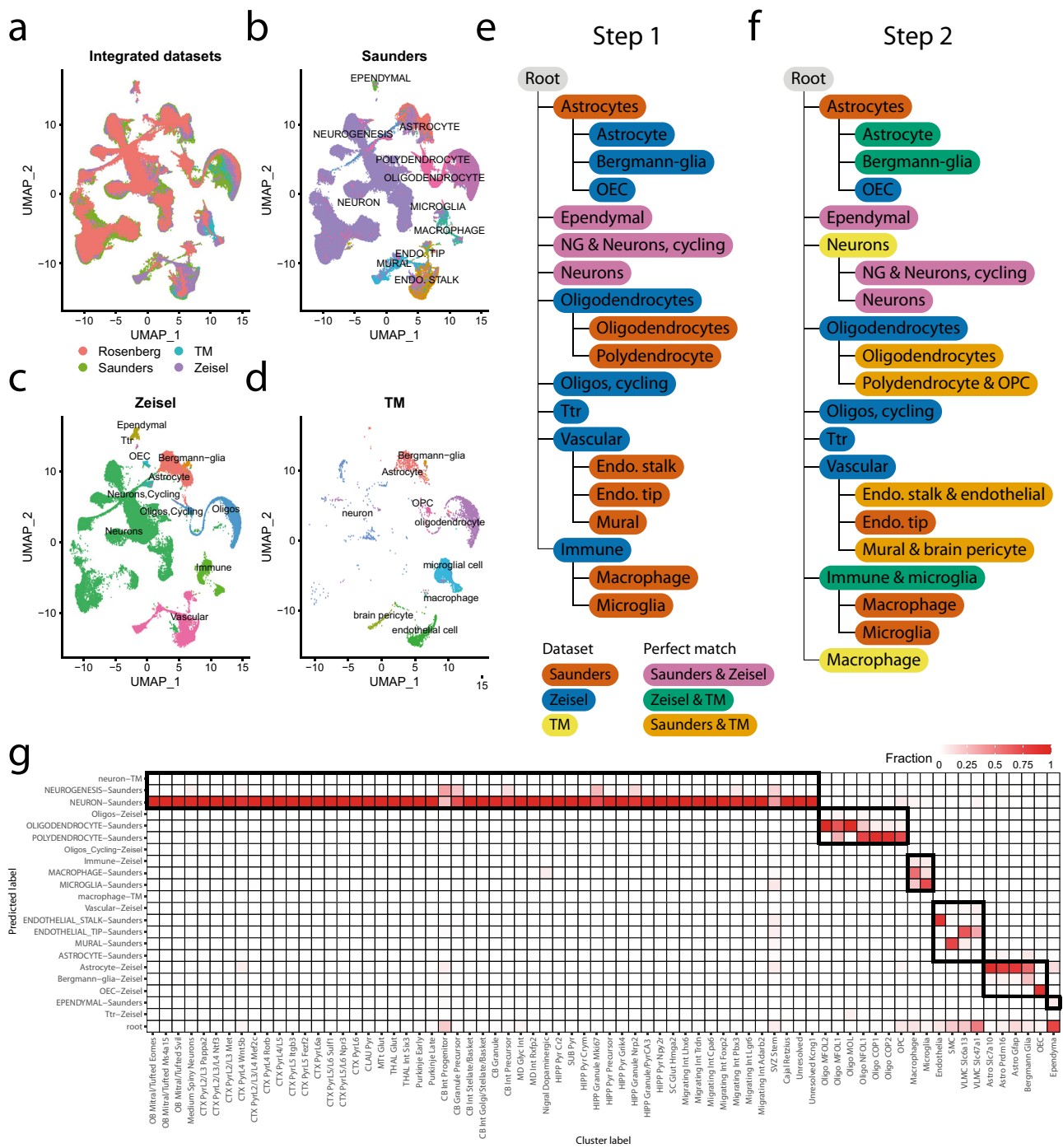

**Fig. 7 Brain inter-dataset evaluation. a–d** UMAP embeddings of the datasets after alignment using Seurat v3. **e** Learned hierarchy when starting with the Saunders dataset and adding Zeisel with linear SVM. **f** Updated tree when the Tabula Muris dataset is added. **g** Confusion matrix when using the learned classification tree to predict the labels of Rosenberg. The dark boundaries indicate the hierarchy of the classification tree.

new relationships between cell populations, such as the mapping of "L6b VISp Col8a1 Rprm" as a subpopulation of "L6b Sla" instead of "L6b Rgs12". This observation would not be possible to make by manually matching populations based on the assigned labels, highlighting the power of automatically constructing cellular hierarchies. In this case, the Cell Ontology database could also not be used to verify this relationship since many brain cell populations are missing. Most of these populations have recently been annotated using scRNA-seq and there is a wide lack of consistency in population annotation and matching between studies[18]. scHPL can potentially be used to map these relations,

irrespective of the assigned labels, and improve the Cell Ontology database.

When combining multiple datasets to construct a tree, we expect that cell populations are annotated correctly. However, in the PBMC inter-dataset experiment, this was not the case. At first sight, the constructed tree looked erroneous, but the expression of marker genes revealed that (parts of) several cell populations were mislabeled. Here, we could use the constructed tree as a warning that there was something wrong with the original annotations.

In general, scHPL is robust to sampling differences between datasets or varying parameters such as the matching threshold or

the number of genes used. The brain datasets used to construct the tree, for instance, varied greatly in population sizes, which did not cause any difficulties. This is mainly because we rely on reciprocal classification. A match between cell populations that is missed when training a classifier on one dataset to predict labels of the other, can still be captured by the classifier trained on the other dataset.

Since batch effects are inevitable when combining datasets, we require datasets to be aligned before running scHPL. In all inter-dataset experiments in this manuscript, we used Seurat V3[28] for the alignment, but we would like to emphasize that scHPL is not dependent on Seurat and can be combined with any batch correction tool, such as more computationally efficient methods like Harmony[33]. A current limitation of these tools is that when a new dataset is added, the alignment—and consequently also scHPL—has to be rerun. An interesting alternative would be to project the new dataset to a latent space learned using reference dataset(s), using scArches[34] for example. In that case, scHPL does not have to be rerun but can be progressively updated.

The batch effects between the datasets make it more difficult to troubleshoot errors. Generally, it will be hard to resolve whether mistakes in the constructed tree are caused by the erroneous alignment of datasets or by mismatches created by scHPL.

We would like to note though that there are inherent limitations to the assumption that cell populations have hierarchical relationships. While this assumption is widely adopted in single-cell studies as well as the Cell Ontology, there are indeed situations in which a tree is not adequate. For instance, situations in which cells dedifferentiate into other cell types, such as beta to alpha cell conversions in type2 diabetes[35,36].

Considering the classification performance, we showed that using a hierarchical approach outperforms flat classification. On the AMB dataset, the linear SVM outperformed SVM_{rejection}, which was the best performing classifier on this dataset[23]. In contrast to SVM_{rejection}, the linear SVM did not reject any of the cells but labeled them as an intermediate cell population. During this experiment, there were no cells of unknown populations. Correct intermediate predictions instead of rejection are therefore beneficial since it provides the user with at least some information. When comparing the linear SVM and one-class SVM, we noticed that the accuracy of the linear SVM is equal to or higher than the one-class SVM on all datasets. For both classifiers, we saw a decrease in performance on populations with a small number of cells, but for the one-class SVM this effect was more apparent.

Since the one-class SVM has a low performance on small cell populations, it also cannot be used to combine datasets that consist of small populations. If the classification performance is low, it will also not be possible to construct the correct tree. On the other hand, the performance of the linear SVM seems to be robust to small populations throughout our experiments. This classifier can thus better be used when combining multiple datasets with small populations.

When testing the rejection option, the one-class SVM clearly outperforms the linear SVM by showing a perfect performance on the simulated dataset. Moreover, when cell populations are missing from the simulated data, the linear SVM cannot learn the correct tree anymore, in contrast to the one-class SVM. This suggests that the one-class SVM is preferred when cell populations are missing, although, on the AMB dataset, the rejection option of both classifiers was not perfect.

In summary, we present a hierarchical progressive learning approach to automatically identify cell identities based on multiple datasets with various levels of subpopulations. We show that we can accurately learn cell identities and learn hierarchical relations between cell populations. Our results indicate that choosing between

a one-class and a linear SVM is a trade-off between achieving higher accuracy and the ability to discover new cell populations. Our approach can be beneficial in single-cell studies where a comprehensive reference atlas is not present, for instance, to annotate datasets consistently during a cohort study. The first available annotated datasets can be used to build the hierarchical tree, which could subsequently be used to annotate cells in the other datasets.

## Methods

**Hierarchical progressive learning**. Within scHPL, we use a hierarchical classifier instead of a flat classifier. A flat classifier is a classifier that does not consider a hierarchy and distinguishes between all cell populations simultaneously. For the AMB dataset, a flat classifier will have to learn the decision boundaries between all 92 cell populations in one go. Alternatively, a hierarchical classifier divides the problem into smaller subproblems. First, it learns the difference between the three broad classes: e.g., GABAergic neurons, glutamatergic neurons, and nonneuronal cells. Next, it learns the decision boundaries between the six subtypes of GABAergic neurons and the eight subtypes of glutamatergic neurons, separately. Finally, it will learn the decision boundaries between the different cell populations within each subtype separately.

**Training the hierarchical classifier**. The training procedure of the hierarchical classifier is the same for every tree: we train a local classifier for each node except the root. This local classifier is either a one-class SVM or a linear SVM. We used the one-class SVM (svm.OneClassSVM(nu = 0.05)) from the scikit-learn library in Python[37]. A one-class classifier only uses positive training samples. Positive training samples include cells from the node itself and all its child nodes. To avoid overfitting, we select the first 100 principal components (PCs) of the training data. Next, we select informative PCs for each node separately using a two-sided two-sample $t$ test between the positive and negative samples of a node ($\alpha < 0.05$, Bonferroni corrected). Negative samples are selected using the siblings policy[38], i.e., sibling nodes include all nodes that have the same ancestor, excluding the ancestor itself. If a node has no siblings, cells labeled as the parent node, but not the node itself are considered negative samples. In some rare cases, the Bonferroni correction was too strict and no PCs were selected. In those cases, the five PCs with the smallest $p$ values were selected. For the linear SVM, we used the svm.LinearSVC() function from the scikit-learn library. This classifier is trained using positive and negative samples. The linear SVM applies L2-regularization by default, so no extra measures to prevent overtraining were necessary.

**The reconstruction error**. The reconstruction error is used to reject unknown cell populations. We use the training data to learn a suitable threshold that can be used to reject cells by doing nested fivefold cross-validation. A PCA (n_components = 100) is learned on the training data. The test data is then reconstructed by first mapping the data to the selected PCA domain, and then mapping the data back to the original space using the inverse transformation (hence the data lies within the plane spanned by the selected PCs). The reconstruction error is the difference between the original data and the reconstructed data (in other words, the distance of the original data to the PC plane). The median of the $q$th (default $q = 0.99$) percentile of the errors across the test data is used as a threshold. By increasing or decreasing this parameter, the number of false negatives can be controlled. Finally, we apply a PCA (n_components = 100) to the whole dataset to learn the transformation that can be applied to new unlabeled data later.

**Predicting the labels**. First, we look at the reconstruction error of a new cell to determine whether it should be rejected. If the reconstruction error is higher than the threshold determined on the training data, the cell is rejected. If not, we continue with predicting its label. We start at the root node, which we denote as the parent node, and use the local classifiers of its children to predict the label of the cell using the predict() function, and score it using the decision_function(), both from the scikit-learn package. These scores represent the signed distance of a cell to the decision boundary. When comparing the results of the local classifiers, we distinguish three scenarios:

1. All child nodes label the cell negative. If the parent node is the root, the new cell is rejected. Otherwise, we have an internal node prediction and the new cell is labeled with the name of the parent node.
2. One child node labels the cell positive. If this child node is a leaf node, the sample is labeled with the name of this node. Otherwise, this node becomes the new parent and we continue with its children.
3. Multiple child nodes label the cell positive. We only consider the child node with the highest score and continue as in scenario two.

**Reciprocal matching labels and updating the tree**. Starting with two datasets, D1 and D2, and the two corresponding classification trees (which can be either hierarchical or flat), we would like to match the labels of the datasets and merge the classification trees accordingly into a new classification tree while being consistent

with both input classification trees (Fig. 1). We do this in two steps: first matching the labels between the two datasets and then updating the tree.

Reciprocal matching labels: We first cross-predict the labels of the datasets: we use the classifier trained on D1 to predict the labels of D2 and vice versa. We construct confusion matrices, C1 and C2, for D1 and D2, respectively. Here, $C1_{ij}$ indicates how many cells of population $i$ of D1 are predicted to be population $j$ of D2. This prediction can be either a leaf node, internal node, or a rejection. As the values in C1 and C2 are highly dependent on the size of a cell population, we normalize the rows such that the sum of every row is one, now indicating the fraction of cells of population $i$ in D1 that has been assigned to population $j$ in D2

$$NC1_{ij} = \frac{C1_{ij}}{\sum_{\forall j} C1_{ij}} \qquad (1)$$

Clearly, a high fraction is indicative of matching population $i$ in D1 with population $j$ in D2. Due to splitting, merging, or new populations between both datasets, multiple relatively high fractions can occur (e.g., if a population $i$ is split in two populations $j_1$ and $j_2$ due to D2 being of a higher resolution, both fractions $NC_{ij1}$ and $NC_{ij2}$ will be approximately 0.5). To accommodate for these operations, we allow multiple matches per population.

To convert these fractions into matches, NC1 and NC2 are converted into binary confusion matrices, BC1 and BC2, where a 1 indicates a match between a population in D1 with a population in D2, and vice versa. To determine a match, we take the value of the fraction as well as the difference with the other fractions into account. This is done for each row (population) of NC1 and NC2 separately. When considering row $i$ from NC1, we first rank all fractions, then the highest fraction will be set to 1 in BC1, after which all fractions for which the difference with the preceding (higher) fraction is less than a predefined threshold (default = 0.25) will also be set to 1 in BC1.

To arrive at reciprocal matching between D1 and D2, we combine BC1 and BC2 into matching matrix X (Fig. 2)

$$X = BC1^T + BC2 \qquad (2)$$

The columns in X represent the cell populations of D1 and the rows represent the cell populations of D2. If $X_{ij} = 2$, this indicates a reciprocal match between cell population $i$ from D2 and cell populations $j$ from D1. $X_{ij} = 1$ indicates a one-sided match, and $X_{ij} = 0$ represents no match.

Tree updating: Using the reciprocal matches between D1 and D2 represented in X, we update the hierarchical tree belonging to D1 to incorporate the labels and tree structure of D2. We do that by handling the correspondences in X elementwise. For a nonzero value in X, we check whether there are other nonzero values in the corresponding row and column to identify which tree operation we need to take (such as split/merge/create). As an example, if we encounter a split for population $i$ in D1 into $j_1$ and $j_2$, we will create new nodes for $j_1$ and $j_2$ as child nodes of node $i$ in the hierarchical tree of D1. Figure 2 and Supplementary Table 1 explain the four most common scenarios: a perfect match, splitting nodes, merging nodes, and a new population. All other scenarios are explained in Supplementary Note 1. After an update, the corresponding values in X are set to zero and we continue with the next nonzero element of X. If the matching is impossible, the corresponding values in X are thus not set to zero. If we have evaluated all elements of X, and there are still non-zero values, we will change X into a strict matrix, i.e., we further only consider reciprocal matches, so all "1"s are turned into a "0" with some exceptions (Supplementary Note 2). We then again evaluate X element-wise once more.

### Evaluation

*Hierarchical F1-score.* We use the hierarchical F1-score (HF1-score) to evaluate the performance of the classifiers[39]. We first calculate the hierarchical precision (hP) and recall (hR)

$$hP = \frac{\sum_i P_i \cap T_i}{\sum_i P_i} \qquad (3)$$

$$hR = \frac{\sum_i P_i \cap T_i}{\sum_i T_i} \qquad (4)$$

Here, $P_i$ is a set that contains the predicted cell population for a cell $i$ and all the ancestors of that node, $T_i$ contains the true cell population and all its ancestors, and $P_i \cap T_i$ is the overlap between these two sets. The HF1-score is the harmonic mean of hP and hR

$$HF1 = \frac{2hP * hR}{hP + hR} \qquad (5)$$

*Median F1-score.* We use the median F1-score to compare the classification performance to other methods. The F1-score is calculated for each cell population in the dataset and afterward the median of these scores is taken. Rejected cells and internal predictions are not considered when calculating this score.

### Datasets

*Simulated data.* We used the R-package Splatter (V 1.6.1) to simulate a hierarchical scRNA-seq dataset that consists of 8839 cells and 9000 genes and represents the tree shown in Supplementary Fig. 1A (Supplementary Note 3)[26]. We chose this low number of genes to speed up the computation time. In total there are 6 different cell populations of approximately 1500 cells each. As a preprocessing step, we log-transformed the count matrix ($\log_2(\text{count}+1)$). A UMAP embedding of the simulated dataset shows it indeed represents the desired hierarchy (Supplementary Fig. 1C).

*Peripheral blood mononuclear cells (PBMC) scRNA-seq datasets.* We used four different PBMC datasets: PBMC-FACS, PBMC-Bench10Xv2, PBMC-Bench10Xv3, and PBMC-eQTL. The PBMC-FACS dataset is the downsampled FACS-sorted PBMC dataset from Zheng et al.[27]. Cells were first FACS-sorted into ten different cell populations (CD14+ monocytes, CD19+ B cells, CD34+ cells, CD4+ helper T cells, CD4+/CD25+ regulatory T cells, CD4+/CD45RA+/CD25− naive T cells, CD4+/CD45RO+ memory T cells, CD56+ natural killer cells, CD8+ cytotoxic T cells, CD8+/CD45RA+ naive cytotoxic T cells) and sequenced using 10× chromium[27]. Each cell population consists of 2000 cells. The total dataset consists of 20,000 cells and 21,952 genes. During the cross-validation on the PBMC-FACS dataset, we tested the effect of selecting HVG. We used the "seurat_v3" flavor of scanpy to select 500, 1000, 2000, and 5000 HVG on the training set[28,40]. The PBMC-Bench10Xv2 and PBMC-Bench10Xv3 datasets are the PbmcBench pbmc1.10Xv2 and pbmc1.10Xv3 datasets from Ding et al.[41]. These datasets consist of 6444 and 3222 cells respectively, 22,280 genes, and nine different cell populations. Originally the PBMC-Bench10Xv2 dataset contained CD14+ and CD16+ monocytes. We merged these into one population called monocytes to introduce a different annotation level compared to the other PBMC datasets. The PBMC-eQTL dataset was sequenced using 10× Chromium and consists of 24,439 cells, 22,229 genes, and eleven different cell populations[42].

*Brain scRNA-seq datasets.* We used two datasets from the mouse brain, AMB2016, and AMB2018, to look at different resolutions of cell populations in the primary mouse visual cortex. The AMB2016 dataset was sequenced using SMARTer[4], downloaded from https://portal.brain-map.org/atlases-and-data/rnaseq/data-files-2018. AMB2016 consists of 1298 cells and 21,413 genes. The AMB2018 dataset, which was sequenced using SMART-Seq V4[5], downloaded from https://portal.brain-map.org/atlases-and-data/rnaseq/mouse-v1-and-alm-smart-seq, consists of 12,771 cells and 42,625 genes. In addition, we used four other brain datasets: Zeisel[2], Tabula Muris[30], Rosenberg[32], and Saunders[31]. These were downloaded from the scArches "data" Google Drive ("mouse_brain_regions.h5ad" from https://drive.google.com/drive/folders/1QQXDuUjKG8CTnwWW_u83MDtdrBXr8Kpq)[34]. We downsampled each dataset such that at the highest resolution each cell population consisted of up to 5000 cells to reduce the computational time for the alignment (Supplementary Table 7).

*Preprocessing scRNA-seq datasets.* All datasets were preprocessed as described in Abdelaal et al.[23]. Briefly, we removed cells labeled in the original studies as doublets, debris or unlabeled cells, cells from cell populations with less than ten cells, and genes that were not expressed. Next, we calculated the median number of detected genes per cell, and from that, we obtained the median absolute deviation (MAD) across all cells in the log scale. We removed cells when the total number of detected genes was below three MAD from the median number of detected genes per cell. During the intra-dataset experiments, we log-transformed the count matrices ($\log_2(\text{count} + 1)$).

*Aligning scRNA-seq datasets.* During the inter-dataset experiments, we aligned the datasets using Seurat V3[28] based on the joint set of genes expressed in all datasets. In the PBMC, AMB, and brain inter-dataset experiment respectively 17,573, 19,197, and 14,858 genes remained. For the PBMC inter-dataset experiment, we also removed cell populations that consisted of less than 100 cells from the datasets used for constructing and training the classification tree (PBMC-eQTL, FACS, Bench10Xv2). To test the effect of the number of genes on scHPL, we integrated this data using 1000, 2000 (default), and 5000 HVGs.

**Reporting summary.** Further information on research design is available in the Nature Research Reporting Summary linked to this article.

### Data availability

The filtered PBMC-FACS and AMB2018 datasets can be downloaded from Zenodo (https://doi.org/10.5281/zenodo.3357167). The simulated dataset and the aligned datasets used during the inter-dataset experiment can be downloaded from Zenodo (https://doi.org/10.5281/zenodo.3736493). Accession numbers or links to the raw data: AMB2016[4] (GSE71585), AMB2018[5] (GSE115746), PBMC-FACS[27] (SRP073767, https://support.10xgenomics.com/single-cell-gene-expression/datasets), PBMC-eQTL[42] (EGAS00001002560), PBMC-Bench10Xv2 and PBMC-Bench10Xv3[41] (GSE132044), Rosenberg[32] (GSE110823), Zeisel[2] (http://mousebrain.org, file name L5_all.loom, downloaded on 9/9/2019), Saunders[31] (http://dropviz.org, DGE by Region section, downloaded on 30/8/2019), Tabula Muris[30] (https://figshare.com/projects/Tabula_Muris_Transcriptomic_characterization_of_20_organs_and_tissues_from_Mus_musculus_at_single_cell_resolution/27733, downloaded on 14/2/2019).

### Code availability

The source code for scHPL is available as a python package that is installable through the PyPI repository (https://github.com/lcmmichielsen/scHPL)[43].

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

## Acknowledgements

This research was supported by an NWO Gravitation project: BRAINSCAPES: A Roadmap from Neurogenetics to Neurobiology (NWO: 024.004.012) and the European Union's Horizon 2020 research and innovation program under the Marie Skłodowska-Curie grant agreement No. 861190 (PAVE).

## Author contributions

L.M., M.J.T.R., and A.M. conceived the study and designed the experiments. L.M. performed all the experiments and wrote the paper. L.M., M.J.T.R., and A.M. reviewed and approved the paper.

## Competing interests

The authors declare no competing interests.
