## [Peer Review File · Nature Communications]

REVIEWER COMMENTS

Reviewer #1 (Remarks to the Author):

Overview

The authors present a novel method, called scHPL, for performing hierarchical classification of cell types in single-cell RNA-seq data. The most novel aspect of this method is its ability to learn a cell type hierarchy from multiple datasets sequentially. Specifically, when presented with a new training set that may have labels representing cell types at different levels of granularity than cell types in previous datasets, scHPL automatically infers the hierarchical relationships between the new cell type labels and the previously seen cell type labels. Moreover, scHPL is able to determine when a cell type in a new dataset has never been present in any previous datasets.

Overall, this manuscript is well written and the method is well described. Furthermore, the authors do a great job in analyzing, in isolation, the various aspects of their algorithm. My biggest concern regarding this manuscript is that it is not clear to me that the task of learning a hierarchy between cell type labels is a problem that requires a computational solution. Said differently, couldn't a researcher manually construct a hierarchy from these disparate datasets using prior knowledge about cell types? The authors need to provide more detailed examples in which this is not possible. Please see my detailed concerns below.

Major Issues:

- My biggest concern regarding this manuscript is that it is not clear to me that the task of learning a hierarchy between cell type labels is a problem that requires a computational solution. That is, couldn't a researcher manually construct a hierarchy from these disparate datasets using prior knowledge about cell types? For example, given a dataset with cells labelled as "T cell" and another dataset with two populations labelled as "CD4+ T cell" and "CD8+ T cell", couldn't a user simply manually create a hierarchy in which "T cell" is a parent of "CD4+ T cell" and "CD8+ T cell". I would imagine that even for more subtle, or newly discovered cell types, it would be possible to manually create such a hierarchy based on the known biology of those cell types.

As the authors mention, there also exists the Cell Ontology, which contains a very rich manually constructed hierarchy of cell types. The authors do not adequately justify why the Cell Ontology is inadequate. Specifically, they state, "These cell ontologies, however, were not developed for scRNA-seq data specifically. As a consequence, many new discovered smaller (sub)populations might be missing and relationships between cell populations might be inaccurate." The authors should provide specific examples of cell type hierarchies that are not captured by the Cell Ontology.

One way to address my concern here would be to provide an example of a set of datasets where the hierarchy of cell types in those datasets are unknown. With such an example, the authors should run scHPL and explore the hierarchy that it constructs. Do the results reveal new biology by describing previously unknown hierarchical relationships between cell types? Such an analysis would demonstrate the utility of scHPL. Without such a demonstration it is difficult to understand when this method would be required.

- In line 213, the authors state, "Moreover, when relying on the predefined hematopoietic tree, CD4+ memory T-cells, CD8+ T-cells, and CD8+ naive T-cells were also often confused, indicating that the learned PBMC tree might better reflect the data." Are the authors claiming that these very established hierarchies should be re-defined? This is a major claim that would need substantially more justification than is provided here.

- I was somewhat confused about the details regarding the simulated datasets whose description starts on line 430. Specifically, the authors state that each of the three "sub"-datasets consisted of 3,000 genes, which were then stacked "column wise" in order to form a dataset with 9,000 genes. Why would each subdataset measure completely disjoint sets of genes? Moreover, the authors state that, "The final dataset consists of 8,839 cells." Why wouldn't the final dataset consist of 18,000 cells from taking the union of the three sub-datasets, each consisting of 9,000 cells? In general, this procedure needs to be described in more detail, with greater clarity.

- Regarding the simulated data, the authors state that the simulated data consisted of only 9,000 genes; however, this is much smaller than the ~20,000 protein coding genes usually measured in a single-cell experiment. The authors should either simulate a more realistic number of genes, or should justify why they chose to decrease the number of genes from what would be expected in real data. It seems that the fewer the number of genes, the easier this task may be, and thus, this simulated data may be easier to classify than real data simply due to the fewer dimensions describing the data.

- In line 210, the authors claim that, "A t-SNE plot of the PBMC-FACS dataset confirms that CD4+ memory T-cells are more similar to CD8+ than CD4+ T-cells based on their transcriptomic profile (Figure S1B)"; however, it is known that t-SNE does not capture global structure in data, only local structure (Wattenberg, et al., "How to Use t-SNE Effectively", Distill). Thus, conclusions should not be drawn from looking at a t-SNE plot. The authors should either consider UMAP instead, given that it has been shown to better preserve global structure (Becht et al., "Dimensionality reduction for visualizing single-cell data using UMAP", Nature Biotechnology), or better yet, use a more rigorous analysis using correlation between the mean expression profiles of these subpopulations.

Minor Issues:

- In line 57, "cell ontology", should be capitalized as "Cell Ontology". It would also benefit to reference Bard, Rhee, and Ashburner (2005) to point readers to this very relevant prior work on attempting to define a cell type hierarchy.

- In line 79, the authors state, "One of these advantages is that a flat classifier needs to distinguish between many classes, while if we exploit the hierarchy, the classification problem is divided into smaller sub-problems." Please expound on what is meant by this. Is there a reference to a specific methodology?

- In Figure 6E, some of the cells of the confusion matrix are outlined in a dark boundary; however, I don't see an explanation for this boundary. The authors should describe these boundaries in the figure caption.

Reviewer #2 (Remarks to the Author):

This manuscript proposes schPL, a novel approach for cell type annotation that combines unstandardized annotations across multiple datasets into a single cell type hierarchy. The principle is similar to progressive learning: the hierarchy starts with a flat tree containing the annotations of the first dataset, but terms of this initial tree are progressively refined as new datasets are iteratively added. One of the main contribution of this paper is the automatic matching and splitting of cell types through classifier methods (as opposed to name matching or mapping to a cell type ontology). The authors propose 4 scenarios that are automatically recognized by the method, notably the capacity to detect cell types that are split into two subtypes in another dataset. Another contribution of the paper is a new rejection method based on a one-class SVM to avoid annotating unseen cell types.

While cluster matching and cross-dataset training are outstanding questions in the field, the manuscript has two critical limitations: (a) it does not handle one of the most interesting scenarios, where clusters only partially overlap (n:m matching), (b) it relies on thresholds that are unlikely to behave well with an increasing number of training datasets and generalize outside of use cases shown in the manuscript.

Major

1. Because partially overlapping clusters are ignored by the procedure ("impossible scenario"), conflicts are likely to arise and propagate throughout the progressive training process, leading to a proliferation of nodes and leaves in the cell type hierarchy. Once errors are introduced, they keep

propagating and lead to internal conflicts in the representation. For example, NK cells are present twice in the cell type hierarchy in Fig. 6B, with “NK (FACS)” inferred as a child of CD8+ cells. The authors state that the UMAP embedding supports that latter conclusion, but on the UMAP, “NK (FACS)” does not seem to be fully contained in the CD8+ clusters, rather there seems to be a partial overlap between NK and CD8+ clusters. In general, scHPL should be able to handle n:m matches for the method to become applicable for a wider audience and most realistic scenarios.

2. scHPL relies on simple ideas (SVM, thresholding, reciprocal matching), but it is unclear that it is robust to noise from batch effects or sampling differences. The use cases presented in the paper are insufficient to show the generalizability of the approach. In a previous benchmark, the authors included difficult annotation tasks, such as predictions across single-cell technologies or across brain regions. In contrast, the present manuscript includes a simulated benchmark, internal predictions on a brain dataset, and a cross-dataset benchmark on PBMC datasets that all use the 10x technology. The manuscript should include more complex cross-dataset use cases to validate their method. In particular, the threshold used to identify matching clusters (0.25) is one of the key parameters of the model, yet no elements are provided about its generalizability, or how it could be adapted for different use cases. When I applied scHPL to a positive control consisting of two datasets with identical cell types, the method did not find any relevant match (it returned a mostly flat hierarchy with a couple of aberrant matches, suggesting that scHPL is not trivially generalizable).

3. Related to the previous point about robustness to batch effects and sampling: on line 263, the authors state that “CD16+ monocytes are predicted to be mDCs and vice versa, which could be explained by the aforementioned fact that monocytes can differentiate into dendritic cells”. Monocytes and DCs being related would explain imperfect classification (mixed CD16+/mDCs predictions), but not why the predictions are perfectly reversed (Fig. 6E), which rather suggests that the classifiers do not generalize.

4. The authors use a pre-defined hierarchy for the Allen Mouse Brain (AMB) dataset, but they do not justify this choice. Even if there is no other dataset to combine, scHPL could have been used on the CV folds. This would act as a positive control in the presence of a large number of cell types: how well can scHPL match cell types that are known to match a priori (in a slightly ideal scenario, as stochastic sampling would be the only source of noise)? In general, the manuscript should provide guidelines about the resolution at which scHPL is expected to work, i.e. how many cell types/what level of heterogeneity can it handle?

5. According to the methods, the authors do not select highly variable genes (HVG) before applying PCA. This is a common step in scRNAseq to filter out noisy components and obtain more generalizable classifiers. This is likely to inflate the number of cases where no matches can be found, which would result in a flat and uninformative cell type hierarchy. When I tested the method, I found that the method was extremely sensitive to input genes. The authors should clarify why they omitted this common preprocessing step.

Minor

- a. The method is implemented as a list of scripts (harder to use than a package) and is insufficiently documented. What input formats are permissible? Is the method compatible with the single-cell ecosystem (AnnData, loom files)? Do you need to preprocess the data? In the vignette, it is never stated that genes need to be identical across datasets, which becomes a problem when you realize that you cannot make predictions on the test dataset (except by manually filtering genes and re-running the vignette).
- b. The methods contain insufficient details on how classifiers are (re)trained once the cell type tree is updated. What happens when an internal node is missing in one of the training datasets? When an internal node has a single child, what classes are being compared?
- c. There seems to be a typo in the hierarchical-F1 definition (l. 415), as the numerators are larger than the denominators for both hP and hR (when both values are ≤ 1). To match the original definition, the union should be an intersection.

Reviewer #1

Overview

The authors present a novel method, called scHPL, for performing hierarchical classification of cell types in single-cell RNA-seq data. The most novel aspect of this method is its ability to learn a cell type hierarchy from multiple datasets sequentially. Specifically, when presented with a new training set that may have labels representing cell types at different levels of granularity than cell types in previous datasets, scHPL automatically infers the hierarchical relationships between the new cell type labels and the previously seen cell type labels. Moreover, scHPL is able to determine when a cell type in a new dataset has never been present in any previous datasets.

Overall, this manuscript is well written and the method is well described. Furthermore, the authors do a great job in analyzing, in isolation, the various aspects of their algorithm. My biggest concern regarding this manuscript is that it is not clear to me that the task of learning a hierarchy between cell type labels is a problem that requires a computational solution. Said differently, couldn't a researcher manually construct a hierarchy from these disparate datasets using prior knowledge about cell types? The authors need to provide more detailed examples in which this is not possible. Please see my detailed concerns below.

Major Issues:

1. My biggest concern regarding this manuscript is that it is not clear to me that the task of learning a hierarchy between cell type labels is a problem that requires a computational solution. That is, couldn't a researcher manually construct a hierarchy from these disparate datasets using prior knowledge about cell types? For example, given a dataset with cells labelled as "T cell" and another dataset with two populations labelled as "CD4+ T cell" and "CD8+ T cell", couldn't a user simply manually create a hierarchy in which "T cell" is a parent of "CD4+ T cell" and "CD8+ T cell". I would imagine that even for more subtle, or newly discovered cell types, it would be possible to manually create such a hierarchy based on the known biology of those cell types. As the authors mention, there also exists the Cell Ontology, which contains a very rich manually constructed hierarchy of cell types. The authors do not adequately justify why the Cell Ontology is inadequate. Specifically, they state, "These cell ontologies, however, were not developed for scRNA-seq data specifically. As a consequence, many new discovered smaller (sub)populations might be missing and relationships between cell populations might be inaccurate." The authors should provide specific examples of cell type hierarchies that are not captured by the Cell Ontology. One way to address my concern here would be to provide an example of a set of datasets where the hierarchy of cell types in those datasets are unknown. With such an example, the authors should run scHPL and explore the hierarchy that it constructs. Do the results reveal new biology by describing previously unknown hierarchical relationships between cell types? Such an analysis would demonstrate the utility of scHPL. Without such a demonstration it is difficult to understand when this method would be required.

Response:

The reviewer raises two important points here which we address separately:

1) Why is the Cell Ontology inadequate?

As we mentioned in the manuscript, the Cell Ontology does not consider the wealth of knowledge arising from scRNA-seq data. A striking example of this inadequacy are neuronal cell populations. Recent single-cell studies have identified hundreds of populations [4, 5, 13, 14], including seven subtypes and 92 cell populations in the dataset of the Allen Mouse Brain alone. In contrast, the Cell Ontology currently includes only one glutamatergic neuronal cell population without any subtypes. This clearly illustrates the difficulty of relying on a curated ontology: it is difficult to keep up with the amount of data resulting from single-cell experiments. We would like to note that the Cell Ontology

can be used and would be particularly powerful in identifying cell populations not seen during training. A recent preprint [12] proposes a method, OnClass, that capitalizes on the Cell Ontology for cell type annotation. While we believe this is a promising direction, relying on the Cell Ontology in its current form significantly limits the ability of these methods since cell populations that are not in the Cell Ontology dataset cannot be predicted nor rejected (a limitation also noted in the OnClass preprint).

2) Can we manually construct hierarchies?

We agree with the reviewer that it is easier to manually construct hierarchical relations in well-studied systems, such as blood immune cells. We would like to note that our choice for PBMCs is indeed because we know these hierarchical relations (to a large extent) and as such can use this system to evaluate our results. However, the practical scenario for scHPL would indeed involve unknown cell types for which defining the tree is not trivial. Nevertheless, we now also show that even in the case of “easily” matchable PBMC populations, relying on assigned labels can be misleading. During the PBMC inter-dataset experiment, we constructed a tree using three datasets. This constructed tree differed from the tree we expected. However, by visualizing marker genes in each dataset separately, we showed that these differences can be explained by (partly) mislabeled cell population (see reviewer 2, point 3). In this situation, manual matching of the cell populations also does not work.

If we consider brain cell populations, the task of manually constructing a cellular hierarchy becomes daunting. While correspondence between major subclasses (e.g. astrocytes, inhibitory neurons, excitatory neurons, microglia, oligodendrocytes) is relatively easy to establish, this is not yet possible for finer subpopulations. There is wide disagreement between neuroscientists on the number of cell populations and how to name them. In a recent paper, Yuste et al. note that a complete, accurate, and precise description of cortical cell types is missing [18]. Without a hierarchy and standardized nomenclature that can be used as a reference (like the one existing for PBMCs), it is impossible to match cell populations between different datasets by name.

To show an example of data in which constructing the tree manually is challenging, we added two new experiments where we construct a tree of brain cell populations.

First, we added an experiment where we combined two scRNA-seq datasets that were sequenced by the Allen Institute for Brain Science in 2016 and 2018 [4, 5]. The 2018 dataset contains more cells (12,771 vs. 1,298 cells) and could thus be annotated at a much higher resolution (92 vs 41 cell populations). We applied scHPL to these datasets and constructed a hierarchical tree. Comparing the resulting tree to the authors’ mapping of cell populations between the 2016 and 2018 studies (Extended Data Figure 6 of Tasic et al. 2018 [5]) showed that scHPL constructed an almost perfect tree using the linear SVM. Only one cell population (‘L6b VISp Col8a1 Rprm’) out of the 92 populations from the 2018 dataset was added to a different 2016 population than expected (‘L6a Sla’ instead of ‘L6b Rsg12’). After visualization of marker genes, however, we noticed that this population expresses *Sla*, but does not express *Rsg12*, which indicate that the mapping obtained with scHPL is probably correct (Figure S18).

Second, we added a more challenging experiment where we construct a tree using three brain datasets sequenced using different technologies by three different labs (Zeisel [2], Saunders [30], and Tabula Muris [29]). After tree construction, we predict the labels of a fourth independent dataset (Rosenberg [31]).

The Tabula Muris dataset contains a cell population called ‘brain pericyte’ and the Zeisel dataset contains a population called ‘Pericytes’. If the datasets were to be mapped manually, these populations would be considered the same and hence merged. However, upon constructing the tree

using scHPL, 'brain pericyte' forms a perfect match with 'Mural (Saunders)' and 'Vascular smooth muscle cells (Zeisel)', while 'pericytes' is a subpopulation of 'Endothelial stalk (Saunders)' and 'Endothelial cell (TM)' (Figure S22-23). Inspection of the UMAP embedding of the integrated datasets also shows that 'pericytes' and 'brain pericyte' are at a different location, but that they do overlap with the cell populations they were matched with (Figure S21). This supports scHPL matching rather than name-based matching. This example clearly illustrates why manual construction of a tree will not always work.

Changes to the manuscript:

- We changed the following section in the *Introduction*
Supervised methods rely either on a reference atlas or labeled dataset. Ideally, we would use a reference atlas containing all possible cell populations to train a classifier. Such an atlas, however, does not exist yet and might never be fully complete. In particular, aberrant cell populations might be missing as a huge number of diseases exist and mutations could result in new cell populations. To overcome these limitations, some methods (e.g. OnClass) rely on the Cell Ontology to identify cell populations that are missing from the training data but do exist in the Cell Ontology database [12]. These Cell Ontologies, however, were not developed for scRNA-seq data specifically. As a consequence, many newly identified (sub)populations are missing and relationships between cell populations might be inaccurate. A striking example of this inadequacy are neuronal cell populations. Recent single-cell studies have identified hundreds of populations [4,13,14], including seven subtypes and 92 cell populations in one study only [5]. In contrast, the Cell Ontology currently includes only one glutamatergic neuronal cell population without any subtypes.
- We added the following section to the *Introduction*
For example, the recently discovered neuronal populations are typically identified using clustering and named based on the expression of marker genes. A standardized nomenclature for these clusters is missing [18], so the relationship between cell populations defined in different datasets is often unknown.
- We added the following section to the *Results* section – *Mapping brain cell populations using scHPL*
Next, we applied scHPL to construct a tree which maps the relationships between brain cell populations. This is a considerably more challenging task compared to PBMCs given the large number of cell populations as well as the fact that brain cell types are not consistently annotated. First, we combined two datasets from the primary visual cortex of the mouse brain, AMB2016 and AMB2018 [4, 5]. AMB2018 contains more cells (12,771 vs. 1,298) and is clustered at a higher resolution (92 cell populations vs. 41) compared to AMB2016. Using scHPL with a linear SVM results in an almost perfect tree (Figure 6). We verified these results by comparing our constructed tree to cluster correspondences in Extended Data Fig. 6 from [5]. Since AMB2018 is clustered at a higher resolution, most populations are subpopulations of AMB2016, which are all correctly identified by scHPL. Conversely, three L4 populations from AMB2016 were merged into one population (L4 IT VISp Rspo1) from AMB2018 [5], forming a continuous spectrum. This relation was also automatically identified using scHPL (Figure 6). Compared to the results from Tasic et al., one cell population from AMB2018 is attached to a different parent node. scHPL assigned 'L6b VISp Col8a1 Rprm' as a subpopulation of 'L6a Sla' instead of 'L6b Rgs12'. This population, however, does not express *Rgs12*, but does express *Sla* (Figure S18), supporting the matching identified by scHPL. Three cell populations could not be added to the tree due to complex scenarios. According to Extended Data Fig. 6 from Tasic et al., these AMB2018 populations are a subpopulation of multiple AMB2016 subpopulations.

The AMB2016 and AMB2018 datasets were generated and analyzed by the same group and hence the cluster matching is certainly easier than a real-life scenario. Therefore, we tested *schPL* also on a complicated scenario with brain datasets that are sequenced using different protocols and by different labs (Table S7). We used three datasets (Zeisel, Tabula Muris, and Saunders) to construct the tree (Figure 7A-D) [2, 29, 30]. The Zeisel dataset is annotated at two resolutions. First, we constructed a tree using a linear SVM based on the low resolution of Zeisel. We started with the Saunders dataset and added Zeisel (Figure 7E). This is a clear illustration of the possible scenarios *schPL* can manage. Some populations are a perfect match between the two datasets (e.g. neurons), some populations from Saunders are splitted (e.g. astrocytes), some are merged (e.g. macrophages and microglia), and some populations from Zeisel have no match (e.g. Ttr). Next, we updated the tree by adding the Tabula Muris dataset (Figure 7F). Here, we found matches that would not have been possible to identify by relying on the assigned cell type labels to map cell types. For example, mural cells from Saunders are a perfect match with the brain pericytes from the Tabula Muris. The results of *schPL* with the one-class SVM were almost identical to the linear SVM (Figure S20A).

Next, we used the resulting tree to predict the labels of a fourth independent dataset (Rosenberg) [31]. The predictions from the linear and the one-class SVM are very similar (Figure 7G, S20B). The only difference is that the linear SVM correctly predicts some progenitor or precursor neuronal populations from Rosenberg to be ‘Neurogenesis’ while the one-class SVM rejects these populations.

To assess the effect of the annotation resolution, we repeated the analysis using the higher resolution annotation from the Zeisel dataset (Figure S21-23). Here, we noticed that the ‘brain pericytes (TM)’ and ‘pericytes (Zeisel)’ - two populations one would easily match based on the names only - are not in the same subtree. ‘Brain pericyte (TM)’ forms a perfect match with ‘Mural (Saunders)’ and ‘Vascular smooth muscle cells (Zeisel)’, while ‘pericytes (Zeisel)’ is a subpopulation of ‘Endothelial stalk (Saunders)’ and ‘Endothelial cell (TM)’ (Figure S22-23). In the UMAP embedding of the integrated datasets, the ‘pericytes’ and ‘brain pericyte’ are at a different location, but they do overlap with the cell populations they were matched with (Figure S21). This highlights the power of *schPL* matching rather than name-based matching.

- We added the following to the *Discussion*:
In this study, we showed that *schPL* can learn cell identities progressively from multiple reference datasets. We showed that using our approach the labels of two AMB datasets can successfully be matched to create a hierarchy containing mainly neuronal cell populations and that we can combine three other brain datasets to create a hierarchy containing mainly non-neuronal cell populations. In both experiments, we discovered new relationships between cell populations, such as the mapping of ‘L6b VISp Col8a1 Rprm’ as a subpopulation of ‘L6b Sla’ instead of ‘L6b Rgs12’. This observation would not be possible to make by manually matching populations based on the assigned labels, highlighting the power of automatically constructing cellular hierarchies. In this case, the Cell Ontology database could also not be used to verify this relationship since many brain cell populations are missing. Most of these populations have recently been annotated using scRNA-seq and there is a wide lack of consistency in population annotation and matching between studies. *schPL* can potentially be used to map these relations, irrespective of the assigned labels, and improve the Cell Ontology database.
- We changed the following section in the *Methods* section – *Brain data*:
We used two datasets from the mouse brain, AMB2016 and AMB2018, data to look at different resolutions of cell populations in the primary mouse visual cortex. The AMB2016 dataset was sequenced using SMARTer [4], downloaded from <https://portal.brain-map.org/atlas-and-data/rnaseq/data-files-2018> and preprocessed as described in [23].

AMB2016 consists of 1,298 cells and 21,413 genes. The AMB2018 dataset, which was sequenced using SMART-Seq V4 [5], downloaded from <https://portal.brain-map.org/atlas-and-data/rnaseq/mouse-v1-and-alm-smart-seq> and preprocessed as described in [19], consists of 12,771 cells and 42,625 genes. For the AMB inter-dataset experiment, only genes expressed in both datasets were considered (19,197 genes). We aligned the two datasets using Seurat V3 [37].

Additionally, we used four other brain datasets: Zeisel [2], Tabula Muris [29], Rosenberg [31], and Saunders [30]. These were downloaded from the scArches 'data' Google drive ('mouse_brain_regions.h5ad' from https://drive.google.com/drive/folders/1QQXDduUjKG8CTnwWW_u83MDtdrBXR8Kpg). We downsampled each dataset such that at the highest resolution each cell population consisted of up to 5,000 cells to reduce the computational time for the alignment (Table S7). We aligned the datasets with Seurat V3 based on the genes expressed in all datasets (14,858 genes) [37].

- We added Figure 6-7, S18-23
- We added Table S7 to the supplement

2. In line 213, the authors state, "Moreover, when relying on the predefined hematopoietic tree, CD4+ memory T-cells, CD8+ T-cells, and CD8+ naive T-cells were also often confused, indicating that the learned PBMC tree might better reflect the data." Are the authors claiming that these very established hierarchies should be re-defined? This is a major claim that would need substantially more justification than is provided here.

Response:

The reviewer is of course right. We definitely do not mean to suggest an alternative hierarchy of PBMCs. What we meant is that the transcriptomic profiles of these cell populations are not sufficient to correctly map their relationships. In this particular example, cell identities were assigned using FACS sorting based on surface protein markers. And even if we provide the "true" tree to the classifier (rather than inferring it using *schPL*), the classifier still cannot correctly predict cell identities. We have now rewritten this part to clarify.

Changes to the manuscript:

- We changed the following section in the *Results: schPL accurately learns cellular hierarchies*. The linear and one-class SVM showed stable results during both experiments; all 60 trees - except for two trees learned by the one-class SVM on the PBMC data - look identical (Figure 4A-D). The final tree for the simulated data looks as expected, but the tree for the PBMC data looks slightly different from the predefined hematopoietic tree (Figure S2A). In the learned trees, CD4+ memory T-cells are a subpopulation of CD8+ instead of CD4+ T-cells. The correlation between the centroids of CD4+ memory T-cell and CD8+ T-cells ($r = 0.985 \pm 0.003$) is also slightly higher than the correlation to CD4+ T-cells ($r = 0.975 \pm 0.002$) (Figure S5). Using the learned tree instead of the predefined hematopoietic tree improves the classification performance of the linear SVM slightly (HF1-score = 0.990 vs 0.985). Moreover, when relying on the predefined hematopoietic tree, CD4+ memory T-cells, CD8+ T-cells, and CD8+ naive T-cells were also often confused, further highlighting the difficulty in distinguishing these populations based on their transcriptomic profiles alone (Tables S4-5).

3. I was somewhat confused about the details regarding the simulated datasets whose description starts on line 430. Specifically, the authors state that each of the three "sub"-datasets consisted of 3,000 genes, which were then stacked "column wise" in order to form a dataset with 9,000 genes.

Why would each subdataset measure completely disjoint sets of genes? Moreover, the authors state that, "The final dataset consists of 8,839 cells." Why wouldn't the final dataset consist of 18,000 cells from taking the union of the three sub-datasets, each consisting of 9,000 cells? In general, this procedure needs to be described in more detail, with greater clarity.

Response:

We thank the reviewer for pointing this out. We have now changed the text such that the simulation experiment is described in a clearer way and added this as a supplementary note.

Changes to the manuscript:

- We changed the following section in the *Methods: Simulated data*
We used the R-package Splatter (v1.6.1) to simulate a hierarchical scRNA-seq dataset that consists of 8,839 cells and 9,000 genes and represents the tree shown in Figure S1A (Supplementary Note 3) [26]. We chose this low number of genes to speed up the computation time. In total there are six different cell populations of approximately 1,500 cells each. As a preprocessing step, we log-transformed the count matrix ($\log_2(count + 1)$). A UMAP embedding of the simulated dataset shows it indeed represents the desired hierarchy (Figure S1C).
- We added the following text to the supplement: *Supplementary Note 3*
Current scRNA-seq data simulators cannot simulate hierarchical data, so we simulated this dataset step by step (Figure S1B).
First, we simulated the expression of 3,000 genes for 9,000 cells. For this simulation, the cells were divided into three groups. The 3,000 simulated genes represent genes that are differentially expressed between the cell populations at a low resolution, so for example B cells vs. T cells. Next, we simulated *another* 3,000 genes for the *same* 9,000 cells. Now, the cells were divided into five groups. Here, the differentially expressed genes represent genes that distinguish cell populations at a slightly higher resolution, so for example CD4+ T cells vs. CD8+ T cells. We repeated this step for another set of 3,000 genes, but now there were six populations. The third dataset represents the highest resolution, so for instance CD4+ memory T cells vs. CD4+ naïve T cells.
Together this resulted in a dataset of 9,000 cells and 9,000 genes. The cells were labeled at three resolutions. There was some inconsistency between the labels at the different resolutions (e.g. some cells were labeled as 'Group12', 'Group3', 'Group3'). We removed these cells from the dataset, which resulted in a final dataset of 8,839 cells and 9,000 genes.
- We moved Figure 3 to the supplement (now Figure S1)

4. Regarding the simulated data, the authors state that the simulated data consisted of only 9,000 genes; however, this is much smaller than the ~20,000 protein coding genes usually measured in a single-cell experiment. The authors should either simulate a more realistic number of genes, or should justify why they chose to decrease the number of genes from what would be expected in real data. It seems that the fewer the number of genes, the easier this task may be, and thus, this simulated data may be easier to classify than real data simply due to the fewer dimensions describing the data.

Response:

We agree with the reviewer that simulating a dataset with 20,000 genes would be more realistic than 9,000 genes. The rational for choosing 9,000 genes was merely to speedup the experiments. We don't think increasing the number of simulated features will have a strong effect on the results since we employ features selection with both classifiers (L2-regularization for the linear SVM and discriminative

PCs for the one-class SVM). Even if the number of features increases, these classifiers can still select the relevant features and be able to distinguish the cell populations.

In another experiment using the PBMC-FACS dataset, we tested the effect of selecting highly variable genes (varying from 500 to 2000) compared to using all genes (see reviewer 2, point 5). There we observed that using a larger number of genes results in a better performance. The performance was highest when all the genes (21,952) are used. For *schPL*, it is thus beneficial to have more genes and rely on the built-in selection scheme.

5. In line 210, the authors claim that, “A t-SNE plot of the PBMC-FACS dataset confirms that CD4+ memory T-cells are more similar to CD8+ than CD4+ T-cells based on their transcriptomic profile (Figure S1B)”; however, it is known that t-SNE does not capture global structure in data, only local structure (Wattenberg, et al., "How to Use t-SNE Effectively", Distill). Thus, conclusions should not be drawn from looking at a t-SNE plot. The authors should either consider UMAP instead, given that it has been shown to better preserve global structure (Becht et al., “Dimensionality reduction for visualizing single-cell data using UMAP”, Nature Biotechnology), or better yet, use a more rigorous analysis using correlation between the mean expression profiles of these subpopulations.

Response:

Following the reviewer’s suggestion, we quantitatively assessed the similarity between cell population using pair-wise Pearson correlation between the population centroids (Figure S5). In general, correlations between all cell populations are high ($r > 0.69$), which again illustrates the difficulty in distinguishing these populations based on the transcriptome. The CD4+ memory T-cells have a slightly higher correlation to the CD8+ T-cells and CD8+ naïve T-cells ($r = 0.985 \pm 0.003$) compared to other CD4+ T-cell populations ($r = 0.975 \pm 0.002$).

Changes to the manuscript:

- We changed the following section in the *Results: schPL accurately learns cellular hierarchies* (see reviewer 1, point 2)
- We added Figure S5

Minor Issues:

a. In line 57, “cell ontology”, should be capitalized as “Cell Ontology”. It would also benefit to reference Bard, Rhee, and Ashburner (2005) to point readers to this very relevant prior work on attempting to define a cell type hierarchy.

Response:

Thanks! We changed this in the manuscript.

b. In line 79, the authors state, “One of these advantages is that a flat classifier needs to distinguish between many classes, while if we exploit the hierarchy, the classification problem is divided into smaller sub-problems.” Please expound on what is meant by this. Is there a reference to a specific methodology?

Response:

This sentence is indeed confusing. We meant here that the hierarchical classifier has an advantage of the flat classifier instead of vice versa.

A flat classifier is a classifier that doesn’t consider a hierarchy. When using a flat classifier, this classifier will have to distinguish between all the cell populations simultaneously. For the AMB dataset, a flat

classifier will have to learn the decision boundaries between all 92 cell populations. A hierarchical classifier divides the problem into smaller subproblems. First it learns the difference between the 3 broad classes: GABAergic neurons, glutamatergic neurons, and non-neuronal cells. Next, it learns the decision boundaries between the six subtypes of GABAergic neurons and the eight subtypes of glutamatergic neurons, separately. Finally, it will learn the decision boundaries between the different cell populations within each subtype separately. When using a hierarchical classifier, the problem is then divided into many smaller sub-problems.

Changes to the manuscript:

- We changed this sentence in the *Introduction*:
One of these advantages is that the classification problem is divided into smaller sub-problems, while a flat classifier needs to distinguish between many classes simultaneously.

- We changed the following section in the *Methods: Hierarchical progressive learning*
Within *schPL*, we use a hierarchical classifier instead of a flat classifier. A flat classifier is a classifier that doesn't consider a hierarchy and distinguishes between all cell populations simultaneously. For the AMB dataset, a flat classifier will have to learn the decision boundaries between all 92 cell populations in one go. Alternatively, a hierarchical classifier divides the problem into smaller subproblems. First it learns the difference between the 3 broad classes: GABAergic neurons, glutamatergic neurons, and non-neuronal cells. Next, it learns the decision boundaries between the six subtypes of GABAergic neurons and the eight subtypes of glutamatergic neurons, separately. Finally, it will learn the decision boundaries between the different cell populations within each subtype separately.

c. In Figure 6E, some of the cells of the confusion matrix are outlined in a dark boundary; however, I don't see an explanation for this boundary. The authors should describe these boundaries in the figure caption.

Response:

These dark boundaries indicate the hierarchy of the constructed tree. We added this to the figure caption.

Reviewer #2 (Remarks to the Author):

This manuscript proposes schPL, a novel approach for cell type annotation that combines unstandardized annotations across multiple datasets into a single cell type hierarchy. The principle is similar to progressive learning: the hierarchy starts with a flat tree containing the annotations of the first dataset, but terms of this initial tree are progressively refined as new datasets are iteratively added. One of the main contribution of this paper is the automatic matching and splitting of cell types through classifier methods (as opposed to name matching or mapping to a cell type ontology). The authors propose 4 scenarios that are automatically recognized by the method, notably the capacity to detect cell types that are split into two subtypes in another dataset. Another contribution of the paper is a new rejection method based on a one-class SVM to avoid annotating unseen cell types.

While cluster matching and cross-dataset training are outstanding questions in the field, the manuscript has two critical limitations: (a) it does not handle one of the most interesting scenarios, where clusters only partially overlap (n:m matching), (b) it relies on thresholds that are unlikely to behave well with an increasing number of training datasets and generalize outside of use cases shown in the manuscript.

Major

1. Because partially overlapping clusters are ignored by the procedure (“impossible scenario”), conflicts are likely to arise and propagate throughout the progressive training process, leading to a proliferation of nodes and leaves in the cell type hierarchy. Once errors are introduced, they keep propagating and lead to internal conflicts in the representation. For example, NK cells are present twice in the cell type hierarchy in Fig. 6B, with “NK (FACS)” inferred as a child of CD8+ cells. The authors state that the UMAP embedding supports that latter conclusion, but on the UMAP, “NK (FACS)” does not seem to be fully contained in the CD8+ clusters, rather there seems to be a partial overlap between NK and CD8+ clusters. In general, schPL should be able to handle n:m matches for the method to become applicable for a wider audience and most realistic scenarios.

Response:

The reviewer points to an important point regarding the handling of partially overlapping scenarios. In schPL, we define a set of logical relationships that the method should be able to handle (e.g. population A in dataset #1 is a subset of population B in dataset #2, etc.). These are described in detail in Supplementary Note 1. There are however other scenarios which we called “impossible scenarios” which we believe are not possible to resolve either manually or computationally. Before we elaborate on those, we would like to note that in case these scenarios occur, the cell populations involved are ignored by schPL (not added to the tree) and as such will not result in errors. We have now also added an option for users to replace a node or add an extra node to the tree if an impossible scenario can be resolved manually.

During a new inter-dataset experiment using two brain datasets from the Allen Institute for Brain Science (see reviewer 1, point 1 for more details), we also saw this n:m matching. Three times, it occurred that there was a population from the AMB2018 dataset that should be added as a subpopulation to multiple populations from the AMB2016 dataset. Based on the reviewer’s comment, we reinvestigated these scenarios. It turned out that one of the matching rules was too strict. With the old rules, too many populations were ignored during an impossible scenario. If population $P1$ and $P2$ were clearly subpopulations of $Q1$, but $P3$ was a subpopulation of $Q1$ and $Q2$ (causing the impossible scenario), all the subpopulations, $P1$, $P2$, and $P3$, were ignored and not added to tree. Now, we changed the rule such that only the population causing the impossible scenario $P3$ is ignored. Thus, for the AMB dataset only the three populations causing this problem are ignored during construction and they also don’t cause an error.

We changed the matching rule such that if population X of one dataset is only predicted to be population Y of the other dataset (so in row X of the binary confusion matrix, there is only one '1'), this match will never be removed in a complex scenario. If population Y has many subpopulations (from which one is complex) and if X is a small population, then the match between X and Y might not be reciprocal since Y is almost never predicted to be X (since X is small compared to its sister populations). Changing this matching rule not only resolved the n:m matches during the AMB inter-dataset experiment, but also greatly improved the results during the PBMC inter-dataset experiment (Figure 5, Figure S16).

With respect to the populations that appear in the tree twice, scHPL relies on the assumption that the input datasets are correctly labeled. If the cells are correctly annotated, it is not possible for NK cells from one dataset to partially overlap with both NK and CD8+ T-cells from the other dataset. Especially during the PBMC inter-dataset experiment we noticed that incorrect annotations lead to mistakes in the tree (see reviewer 2, point 3). In the tree constructed by the one-class SVM, for instance, CD14+ monocytes are added to the tree twice. After visualizing marker genes, we noticed that part of the CD14+ monocytes in the FACS dataset are actually mDC. This explains why this population is not a perfect match with the other CD14+ monocyte population, but instead merges the monocytes and CD16+ monocytes (which are also incorrectly annotated and are actually mDC based on the marker genes, see point 3 for more details).

Finally, we would like to note that there are inherent limitations to the assumption that cell populations have hierarchical relationships. While this assumption is widely adopted (e.g. the Cell Ontology), there are indeed situations in which a tree is not adequate (e.g. diseases in which cells dedifferentiate into other cell types, such as beta to alpha cell conversions in type2 diabetes).

Changes to the manuscript:

- We added two paragraphs to the *Discussion*:
When combining multiple datasets to construct a tree, we expect that cell populations are annotated correctly. However, in the PBMC inter-dataset experiment, this was not the case. At first sight, the constructed tree looked erroneous, but the expression of marker genes revealed that (parts of) several cell populations were mislabeled. Here, we could use the constructed tree as a warning that there was something wrong with the original annotations.

We would like to note though that there are inherent limitations to the assumption that cell populations have hierarchical relationships. While this assumption is widely adopted in single cell studies as well as the Cell Ontology, there are indeed situations in which a tree is not adequate. For instance, situations in which cells dedifferentiate into other cell types, such as beta to alpha cell conversions in type2 diabetes [32, 33].

- We added *Supplementary Note 2* to discuss the exceptions to the reciprocal matching rule:
If there is a complex scenario that cannot be solved immediately, matrix X will be changed into a strict matrix. In the strict matrix, only reciprocal matches are considered, so all '1's' are turned into '0'. There are some exceptions to this rule.
 - A population can never have a reciprocal match with the root, so these '1's' are never removed.
 - If a population from a dataset has only one match, it is also never removed. Consider the following example: If population P1 of Dataset 1 is only predicted to be Population Q of Dataset 2, we know that P1 should be a match with Q as it cannot be matched with any other population or with the root. It could be that this match is not reciprocal if population Q has many different subpopulations (e.g. P1, P2, P3, P4). Imagine that population P2 is really big. Almost all cells of population Q will be predicted to be P2

and so the matches with P1 (and P3 and P4) are missed because of the matching threshold. In case there is a complex scenario caused by any other population (maybe P2 or P3 or P4), we still know that P1 is a subpopulation of Q, since that was super clear and didn't cause any complexity.

2. scHPL relies on simple ideas (SVM, thresholding, reciprocal matching), but it is unclear that it is robust to noise from batch effects or sampling differences. The use cases presented in the paper are insufficient to show the generalizability of the approach. In a previous benchmark, the authors included difficult annotation tasks, such as predictions across single-cell technologies or across brain regions. In contrast, the present manuscript includes a simulated benchmark, internal predictions on a brain dataset, and a cross-dataset benchmark on PBMC datasets that all use the 10x technology. The manuscript should include more complex cross-dataset use cases to validate their method. In particular, the threshold used to identify matching clusters (0.25) is one of the key parameters of the model, yet no elements are provided about its generalizability, or how it could be adapted for different use cases. When I applied scHPL to a positive control consisting of two datasets with identical cell types, the method did not find any relevant match (it returned a mostly flat hierarchy with a couple of aberrant matches, suggesting that scHPL is not trivially generalizable.

Response:

The reviewer raises an important point regarding the robustness of scHPL.

First, regarding the robustness of scHPL to batch effects and sampling differences. Before applying scHPL, we aligned the different datasets using Seurat v3. scHPL is however independent from the alignment step and as such, any other alignment method can be used instead. We have now clarified this further in the manuscript and the vignettes on Github. Regarding the sampling differences, the real datasets tested in the manuscript differ significantly in population sizes so we believe this not to be an issue. During the brain inter-dataset experiment, for instance, the Saunders dataset consists of 389,439 cells and 11 cell populations, while the Tabula Muris consists of only 7,856 cells and 9 cell populations (Table S7). The proportions of the cell populations also vary greatly (Figure S19). In the Saunders dataset, most cells are annotated as neuron (232,815), while in the Tabula Muris dataset there are only 281 neuronal cells.

Second, we added an additional experiment to test the effect of varying the population matching threshold. We varied the matching threshold when constructing the tree for the PBMC-FACS dataset. We tested three different thresholds: 0.1, 0.25, and 0.5. The trees constructed using the linear SVM classifier were identical in all three experiments. For the one-class SVM, only 2 out of the resulting 60 trees changed. In these cases, the link that was missing between the CD8+ T-cells and CD8+ naïve and CD4+ memory cells (Figure 4D), is present in these two trees. The effect of the population matching threshold is minimal due to the fact that scHPL relies on reciprocal classification to match populations. Consider a scenario where there is a population A in dataset 1 that has subpopulations a_1 , a_2 , and a_3 of varying size in dataset 2. When we use the classifier from dataset 1 to predict the labels of dataset 2, all these subpopulations will be predicted to be A and thus have a match with A. Vice versa, population A might be matched to the largest of the subpopulations (assume a_1) but not to a_2 and a_3 , due to the matching threshold. But this will not be a problem, since we already knew there was a match based on the other classification.

Finally, we have added two additional experiments to the revised manuscript to illustrate the generalizability of scHPL to complex brain datasets. In one experiment, we learned the tree from two datasets from the primary visual cortex of the mouse brain. In the second experiment, we applied scHPL to three brain datasets from different labs using different technologies and used the resulting

tree to classify cells from an independent forth dataset. Both experiments illustrate that scHPL can reconstruct cellular hierarchies accurately in complex situations.

Changes to the manuscript:

- We added the following text to the end of the section *Results: scHPL correctly learns cellular hierarchies*
Next, we tested the effect of the matching threshold (default = 0.25) on the tree construction by varying this to 0.1 and 0.5. For the linear SVM, changing the threshold had no effect. For the one-class SVM, we noticed a small difference when changing the threshold to 0.1. The two trees that were different using the default threshold (Figure 4D), were now constructed as the other 58 trees. In general, scHPL is robust to settings of the matching threshold due to its reliance on reciprocal classification.
- We added the following section to the *Results* section – *Mapping brain cell populations using scHPL* (see reviewer 1, point 1)
- We changed the following section in the *Methods* section – *Brain data* (see reviewer 1, point 1)
- We added Figure 6-7, S18-23
- We added Table S7 to the supplement
- We added a paragraph to the *Discussion*
In general, scHPL is robust to sampling differences between datasets or varying parameters such as the matching threshold or the number of genes used. The brain datasets used to construct the tree, for instance, varied greatly in population sizes, which did not cause any difficulties. This is mainly because we rely on reciprocal classification. A match between cell populations that is missed when training a classifier on one dataset to predict labels of the other, can still be captured by the classifier trained on the other dataset.

3. Related to the previous point about robustness to batch effects and sampling: on line 263, the authors state that “CD16+ monocytes are predicted to be mDCs and vice versa, which could be explained by the aforementioned fact that monocytes can differentiate into dendritic cells”. Monocytes and DCs being related would explain imperfect classification (mixed CD16+/mDCs predictions), but not why the predictions are perfectly reversed (Fig. 6E), which rather suggests that the classifiers do not generalize.

Response:

We thank the reviewer for this interesting remark. We visualized the marker genes for each cell population in the datasets separately and realized that some cell populations (or parts of cell populations) were mislabeled in the original publications. We detail these cases here:

1. mDC and CD16+ monocytes are mislabeled in the eQTL dataset (Figure S6-8). mDC express *FCGR3A*, a marker for CD16+ monocytes. The CD16+ monocytes express *FCER1A*, which is an mDC marker gene. This explains the ‘wrong’ predictions on the Bench10Xv3 dataset.
2. Part of the CD14+ monocytes in the FACS dataset express *FCER1A* and thus should be labeled as mDC. This also explains why the mDC (which are called CD16+ MC in the tree) from the eQTL dataset are not attached to the root, but attached as a child node to the CD14+ monocytes from the FACS dataset (Figure S6, S8-9).
3. The CD8+ T-cells in the eQTL, 10Xv2, and 10Xv3 datasets are CD8+ cytotoxic T-cells, while the CD8+ naïve T-cells are mislabeled as CD4+ T-cells (Figure S6, S10-13). This explains why the

CD8+ cytotoxic T-cells from the FACS dataset are a subpopulation of the CD4+ T-cells from the eQTL and Bench10Xv2 dataset in the constructed trees. This also explains why part of the CD4+ T-cells from the Bench10Xv3 dataset is classified as CD8+ naïve T-cells.

4. CD34+ cells from the FACS dataset are partly pDC (Figure S6, S14-15). These cells don't express CD34+ and show low expression of *SERPINF1*, which is a marker gene for pDC. The cells from the FACS dataset are sorted, so you would expect that their label is correct. The purity of the CD34+ cells, however, was really low (45%) [27].

These results further highlight the power of scHPL in accurately matching cells based on their transcriptional similarity rather than the assigned labels. Additionally, scHPL allowed us to identify these mislabeled populations (based on the resulting unrealistic name matchings).

Changes to the manuscript:

- We changed the following section in the *Results: Linear SVM can learn the classification tree during an inter-dataset experiment*

When comparing the tree learned using the linear SVM to the expected tree, we notice five differences (Figure 5A-B). Some of these differences are minor, such as the matching of monocytes from the Bench10Xv2 dataset to myeloid dendritic cells (mDC), CD14+ monocytes, and the CD16+ monocytes. Monocytes can differentiate into mDC which can explain their transcriptomic similarity [28]. Other differences between the reconstructed and the expected trees are likely the result of (partly) mislabeled cell populations in the original datasets (Figure S6-15). (i) According to the expression of *FCER1A* (a marker for mDC) and *FCGR3A* (CD16+ monocytes), the labels of the mDC and the CD16+ monocytes in the eQTL dataset are reversed (Figure S6-8). (ii) Part of the CD14+ monocytes in the FACS dataset express *FCER1A*, which is a marker for mDC (Figure S6, S8-9). The CD14+ monocytes in the FACS dataset are thus partly mDCs, which explains the match with the mDC from the eQTL dataset. (iii) Part of the CD4+ T-cells from the eQTL and Bench10Xv2 dataset should be relabeled as CD8+ T-cells (Figure S6, S10-13). In these datasets, the cells labeled as the CD8+ T-cells only contain cytotoxic CD8+ T-cells, while naïve CD8+ T-cells are all labeled as CD4+ T-cells. This mislabeling explains why the CD8+ naïve T-cells are a subpopulation of the CD4+ T-cells. (iv) Part of the CD34+ cells in the FACS dataset should be relabeled as pDC (Figure S6, S14-15), which explains why the pDC are a subpopulation of the CD34+ cells. In the FACS dataset, the labels were obtained using sorting, which would indicate that these labels are correct. The purity of the CD34+ cells, however, was significantly low (45%) compared to other cell populations (92-100%) [27]. There is one difference, however, that cannot be explained by mislabeling. The NK cells from the FACS dataset do not only match the NK cells from the eQTL dataset, but also the CD8+ T-cells.

Most cells of the Bench10Xv3 dataset can be correctly annotated using the learned classification tree (Figure 5E). Interestingly, we notice that the CD16+ monocytes are predicted to be mDCs and vice versa, which could be explained by the fact that the labels of the mDCs and the CD16+ monocytes were flipped in the eQTL dataset. Furthermore, part of the CD4+ T-cells are predicted to be CD8+ naïve T-cells. In the Bench10Xv3, we noticed the same mislabeling of part of the CD4+ T-cells as in the eQTL and Bench10Xv2 datasets, which supports our predictions (Figure S6, S10-13).

The tree constructed using the one-class SVM differs slightly compared to the linear SVM (Figure S16). Here, the monocytes from the Bench10Xv2 match the CD14+ monocytes and mDC (which are actually CD16+ monocytes) as we would expect. Next, the CD14+ monocytes from the FACS dataset merge the CD16+ monocytes (which are actually mDC) and the monocytes. Using the one-class SVM the CD8+ T-cells and NK cells from the Bench10Xv2 dataset are missing since there was a complex scenario. The NK cells are a relatively small population in this dataset which made it difficult to train a classifier for this population.

- We added Figure S6-15

4. The authors use a pre-defined hierarchy for the Allen Mouse Brain (AMB) dataset, but they do not justify this choice. Even if there is no other dataset to combine, scHPL could have been used on the CV folds. This would act as a positive control in the presence of a large number of cell types: how well can scHPL match cell types that are known to match a priori (in a slightly ideal scenario, as stochastic sampling would be the only source of noise)? In general, the manuscript should provide guidelines about the resolution at which scHPL is expected to work, i.e. how many cell types/what level of heterogeneity can it handle?

Response:

We did indeed not test the tree construction on the AMB dataset. We used a predefined hierarchy such that we could compare the classification performance to the results obtained during the benchmark. We have now added an extra inter-dataset experiment where we construct a hierarchy using the AMB dataset from 2018 and 2016 to test whether scHPL can handle a large number of (small) cell populations (see reviewer 1 point 1). In general, scHPL can handle datasets with a high resolution and high level of heterogeneity. For the linear SVM, the classification performance is not that much affected by the number of populations nor their size. This classifier has thus no problem with constructing the hierarchy during the AMB inter-dataset experiment. On the other hand, the classification performance of the one-class SVM decreases when the size of the cell population decreases (Figure 3F).

Furthermore, we added a second brain inter-dataset where we constructed a tree using the Saunders, Zeisel, and Tabula Muris dataset. The Zeisel dataset is annotated at two different resolutions. The low resolution contains 11 populations, while at the high resolution the dataset contains 30 populations. During these experiments, we show that the linear SVM can construct a correct tree regardless of the resolution of Zeisel used (Figure 7, S22). On the contrary, the one-class SVM suffers when the high resolution is used (Figure S20, S23).

Changes to the manuscript:

- We added the following to the *Discussion*:
Since the one-class SVM has a low performance on small cell populations, it also cannot be used to combine datasets which consist of small populations. If the classification performance is low, it will also not be possible to construct the correct tree. On the other hand, the performance of the linear SVM seems to be robust to small populations throughout our experiments. This classifier can thus better be used when combining multiple datasets with small populations.

5. According to the methods, the authors do not select highly variable genes (HVG) before applying PCA. This is a common step in scRNAseq to filter out noisy components and obtain more generalizable classifiers. This is likely to inflate the number of cases where no matches can be found, which would result in a flat and uninformative cell type hierarchy. When I tested the method, I found that the method was extremely sensitive to input genes. The authors should clarify why they omitted this common preprocessing step.

Response:

The reviewer is right that we did not select HVG since the two classifiers we used select features before or during training. For the one-class SVM we select informative PCs and only use these to train the classifier. The linear SVM selects relevant features by using L2-regularization during training. Because of these feature selection steps, we expect that the effect of selecting HVG would be minimal. To verify this, we added two extra experiments:

1. We tested the effect of HVG selection on the classification performance on the PBMC-FACS dataset during cross validation. We selected 500, 1000, 2000, and 5000 HVG on the training fold using the 'seurat_v3' flavor of scanpy. When comparing the HF1-scores on the test set, we notice that the performance of as well the linear SVM as the one-class SVM increases when the number of genes increases (Figure S4). The best performance was obtained using all the genes, so we prefer not to select HVG.
2. We tested the effect of HVG selection in the PBMC inter-dataset experiment. In this experiment, the datasets were aligned using Seurat V3. Originally, we used the default setting of 2,000 HVGs. We have now integrated the data using 1,000 and 5,000 HVGs to test what the effect of the number of HVGs is on the tree construction and the classification performance on the Bench10Xv3 dataset. Here, we show that the effect of gene selection on the tree construction is minimal. For the one-class SVM, all trees look the same. For the linear SVM, we notice one small difference when using 1000 genes: the CD8+ T-cells from the Bench10Xv2 dataset are a subpopulation of the CD8+ T-cells from the eQTL dataset instead of a perfect match.
The predicted labels of the Bench10Xv3 dataset change slightly when using a different number of features (Figure S17). When using 5,000 HVGs, the classification performance seems a bit better, especially for the megakaryocytes, but it is cell population dependent what is the best.

Changes to the manuscript:

- We changed two sentences to the following section in the *Methods: PBMC data*
We used the 'seurat_v3' flavor of scanpy to select 500, 1000, 2000, and 5000 HVG on the training set [37, 38].

Next, we aligned the datasets using Seurat V3 [37]. To test the effect of the number of genes on scHPL, we integrated the data using 1000, 2000 (default), and 5000 HVGs.
- We added extra text to the section: *Results: Linear SVM has a higher classification accuracy than one-class SVM*
In the previous experiments, we used all genes to train the classifiers. Since the selection of highly variable genes (HVGs) is common in scRNA-seq analysis pipelines, we tested the effect of selecting HVGs on the classification performance of the PBMC-FACS dataset. We noted that using all genes results in the highest HF1-score for both the linear and one-class SVM (Figure S4).
- We added extra text to the section: *Results: Linear SVM can learn the classification tree during an inter-dataset experiment*
In the previous experiments, we used the default setting of Seurat to align the datasets (using 2000 genes). We tested whether changing the number of genes to 1000 and 5000 affects the performance. When using the one-class SVM, the number of genes does not affect tree construction. For the linear SVM, we notice one small difference when using 1000 genes: the CD8+ T-cells from the Bench10Xv2 dataset are a subpopulation of the CD8+ T-cells from the eQTL dataset instead of a perfect match.
The predicted labels of the Bench10Xv3 dataset change slightly when using a different number of genes(Figure S17). Whether more genes improves the prediction, differs per cell population. The labels of the megakaryocytes, for instance, are better predicted when more genes are used, but for the dendritic cells we observe the reverse pattern.
- We added Figure S4, and Figure S17

Minor

a. The method is implemented as a list of scripts (harder to use than a package) and is insufficiently documented. What input formats are permissible? Is the method compatible with the single-cell ecosystem (AnnData, loom files)? Do you need to preprocess the data? In the vignette, it is never stated that genes need to be identical across datasets, which becomes a problem when you realize that you cannot make predictions on the test dataset (except by manually filtering genes and re-running the vignette).

Response:

We thank the reviewer for this valuable remark. We have now implemented scHPL as a python package that is installable through the PyPI repository . We have also updated the Github repository to clarify the data loading, preprocessing, selection of identical genes etc. We also added a vignette where we explain how an AnnData object can be used as an input.

b. The methods contain insufficient details on how classifiers are (re)trained once the cell type tree is updated. What happens when an internal node is missing in one of the training datasets? When an internal node has a single child, what classes are being compared?

Response:

We explained this procedure more clearly now. To train the one-class SVM, only positive samples are used. For the linear SVM both positive and negative samples are used. Positive samples include all cells from the node itself as well as cells from its child nodes. Negative samples are selected using the siblings policy. Siblings include all the nodes with the same ancestor excluding the ancestor itself. If an internal node has no cells (e.g. it could be that all cells in a dataset are subpopulations of CD4+ T-cells, but none are labeled as CD4+ T-cells), this is no problem, since the child nodes (all the CD4+ subpopulations) are also used as positive samples.

When an internal node has only a single child, training a classifier for this child node using the one-class SVM does not change. This classifier only uses positive samples, so the samples of that child node can be used. The linear SVM, however, also needs negative samples. In this case, we use the cells labeled as the child node as the positive samples and the cells labeled as the parent node but not as the child node as negative samples.

Changes to the manuscript:

- We changed the following section in the *Methods: Training the hierarchical classifier*
The training procedure of the hierarchical classifier is the same for every tree: we train a local classifier for each node except the root. This local classifier is either a one-class SVM or a linear SVM. We used the one-class SVM (*svm.OneClassSVM(nu = 0.05)*) from the scikit-learn library in Python [34]. A one-class classifier only uses positive training samples. Positive training samples include cells from the node itself and all its child nodes. To avoid overfitting, we select the first 100 principal components (PCs) of the training data. Next, we select informative PCs for each node separately using a two-sided two-sample t-test between the positive and negative samples of a node ($\alpha < 0.05$, Bonferroni corrected). Negative samples are selected using the siblings policy [35], i.e. sibling nodes include all nodes that have the same ancestor, excluding the ancestor itself. If a node has no siblings, cells labeled as the parent node, but not the node itself are considered negative samples. In some rare cases, the Bonferroni correction was too strict and no PCs were selected. In those cases, the five PCs with the smallest p-values were selected. For the linear SVM, we used the *svm.LinearSVC()* function from the scikit-learn library. This classifier is trained using positive and negative samples. The linear SVM applies L2-regularization by default, so no extra measures to prevent overtraining were necessary.

c. There seems to be a typo in the hierarchical-F1 definition (l. 415), as the numerators are larger than

the denominators for both hP and hR (when both values are ≤ 1). To match the original definition, the union should be an intersection.

Response:

We thank the reviewer for pointing this out and changed this in the manuscript.

REVIEWERS' COMMENTS

Reviewer #1 (Remarks to the Author):

Thanks to the authors for the thorough and helpful responses to my concerns. In my view, the additional experiments that entail running scHPL on brain data for which the hierarchy is not well established was very helpful for demonstrating that scHPL will be useful in practice. It appears that scHPL will be most useful to researchers who wish to combine datasets that have been annotated using novel cell type labels that have not been well standardized. Moreover, by detecting the "impossible scenario" in their tree updating algorithm, scHPL will be helpful for identifying inconsistencies between two sets of cell type labels. Congratulations to the authors on the great work!

Minor comments

- In Supplementary Note 1, the figure numbers appear to be incorrect. For example, I believe "Figure S7" should be "Figure S27".

Reviewer #2 (Remarks to the Author):

scHPL turns a confusion matrix (cell type x cell type) into a consensus taxonomy preserving original author annotations. It works, but is inadequately explained for a methods paper. The necessity to pre-align the datasets limits the utility of the methods.

Positive points after revision:

- The new implementation removes as few populations as possible, which is a clear improvement
- The new datasets provide a good overview of how/when the method is expected to work.

This is a methods paper, requirements and limitations should be clear for potential users. Two points discussed in the rebuttal are still insufficiently clear in the current manuscript:

- Aligning input datasets is required. Currently, the authors only mention that they aligned datasets using Seurat for their specific use cases but never clearly state in the results, the description of the method, or the vignettes that this is a required step for the method to work. The method switches from using dataset-specific classifiers (to generate the consensus tree) to a classifier on the merged data, which could also be made more explicit in the method description.
- Cell populations may be removed from the consensus tree (n:m mapping). It's important for users to know about this limitation, which is currently only clearly stated in the supplementary notes (it could be mentioned in the methods at least).

Having to pre-align datasets reduces scHPL's usability, scalability and extensibility: pre-aligning with Seurat is expensive, so one may want to keep just using Seurat.

Using Seurat, one would realistically perform consensus clustering and use confusion matrices to create a mapping between consensus cell types and original cell types, which is a fuzzy version of what scHPL does (plus it does not remove any cell type from the mapping). Consensus labels would then be derived manually (as reviewer 1 suggested). On the other hand, people may be happy to have a software do the dirty work and have clear-cut decision taken for them.

Two additional issues: hard to incorporate a lot of datasets due to alignment cost, hard to troubleshoot assignment errors (did the problem stem from the alignment or scHPL?).

I think a more modest, focused title should be regarded as a requirement.

REVIEWERS' COMMENTS

Reviewer #1 (Remarks to the Author):

Thanks to the authors for the thorough and helpful responses to my concerns. In my view, the additional experiments that entail running scHPL on brain data for which the hierarchy is not well established was very helpful for demonstrating that scHPL will be useful in practice. It appears that scHPL will be most useful to researchers who wish to combine datasets that have been annotated using novel cell type labels that have not been well standardized. Moreover, by detecting the "impossible scenario" in their tree updating algorithm, scHPL will be helpful for identifying inconsistencies between two sets of cell type labels. Congratulations to the authors on the great work!

Response: Thanks for the nice words.

Minor comments

- In Supplementary Note 1, the figure numbers appear to be incorrect. For example, I believe "Figure S7" should be "Figure S27".

Response: The reviewer is right and we have now changed it.

Reviewer #2 (Remarks to the Author):

scHPL turns a confusion matrix (cell type x cell type) into a consensus taxonomy preserving original author annotations. It works, but is inadequately explained for a methods paper. The necessity to pre-align the datasets limits the utility of the methods.

Positive points after revision:

- The new implementation removes as few populations as possible, which is a clear improvement
- The new datasets provide a good overview of how/when the method is expected to work.

Response: Thanks for noticing these improvements.

This is a methods paper, requirements and limitations should be clear for potential users. Two points discussed in the rebuttal are still insufficiently clear in the current manuscript:

- Aligning input datasets is required. Currently, the authors only mention that they aligned datasets using Seurat for their specific use cases but never clearly state in the results, the description of the method, or the vignettes that this is a required step for the method to work. The method switches from using dataset-specific classifiers (to generate the consensus tree) to a classifier on the merged data, which could also be made more explicit in the method description.

Response:

It is true that scHPL requires the datasets to be aligned with each other. While this was mentioned this in the Methods section, we now explicitly mention this in the Results section, changed the README on GitHub, and discuss this requirement in the Discussion section.

Changed text:

- *In the results:* Before applying scHPL, we aligned the datasets using Seurat²⁸.
- *In discussion section:* Since batch effects are inevitable when combining datasets, we require datasets to be aligned before running scHPL. In all inter-dataset experiments in this manuscript we used Seurat v3²⁸ for the alignment, but we would like to emphasize that scHPL is not dependent on Seurat and can be combined with any batch correction tool, such as more computationally efficient methods like Harmony³³. A current limitation of these tools is that when a new dataset is added, the alignment - and consequently also scHPL - has to be rerun. An interesting alternative would be to project the new dataset to a latent space learned using reference dataset(s), using scArches³⁴ for example. In that case, scHPL does not have to be rerun but can be progressively updated.

• Cell populations may be removed from the consensus tree (n:m mapping). It's important for users to know about this limitation, which is currently only clearly stated in the supplementary notes (it could be mentioned in the methods at least).

Response:

This may happen indeed, but instead of being a limitation, this points to inconsistencies in the labels of the original datasets and could guide the users to correct mislabeled populations. We have shown this in the manuscript when combining the AMB2016 and AMB2018 datasets: three populations from AMB2018 could not be added to the tree, which could be explained by these AMB2018 populations being subpopulations of multiple AMB2016 populations (extended Data Fig. 6 from Tasic et al.) pointing to inconsistent labelling between the AMB2016 and AMB2018 dataset. To clarify this, we now additionally mention in the results section that this can happen and added a warning in the Vignette. We also mention in the Vignettes, that we do provide an option in the implementation of scHPL to keep all populations, even those involved in "impossible scenarios", in which case they are added to the root.

Changed text:

- *In the results, added the following:* It could happen that datasets are inconsistently labeled, and the labels cannot be matched (Supplementary Note 1). In this case, one of the populations might be missing from the tree.

Having to pre-align datasets reduces scHPL's usability, scalability and extensibility: pre-aligning with Seurat is expensive, so one may want to keep just using Seurat.

Response:

We would like to note that while scHPL depends on pre-aligned datasets, scHPL is independent of the chosen alignment method. Some alignment methods such as harmony are known to be computationally very efficient. We mention this now in the discussion section (see earlier answer).

Using Seurat, one would realistically perform consensus clustering and use confusion matrices to create a mapping between consensus cell types and original cell types, which is a fuzzy version of what scHPL does (plus it does not remove any cell type from the mapping). Consensus labels would then be derived manually (as reviewer 1 suggested). On the other hand, people may be happy to have a software do the dirty work and have clear-cut decision taken for them.

Response:

The approach suggested by the reviewer is interesting and could be an alternative research direction optimizing the cell-to-cell mappings. Yet, it still requires a manual step, for which (in our answer to reviewer 1 in the previous rebuttal) we have shown (through multiple examples) that it is not trivial/feasible. Moreover, the proposed direction does not resolve the hierarchical tree

construction, as well as the hierarchical approach to classify cell types, which we believe is a major contribution of scHPL.

Two additional issues: hard to incorporate a lot of datasets due to alignment cost, hard to troubleshoot assignment errors (did the problem stem from the alignment or scHPL?).

hard to incorporate a lot of datasets due to alignment cost

Response:

When moving towards automatic cell type identification, alignment of data sets will be a prerequisite for any method (either explicit or built-in). Next, as scHPL is independent of the alignment method, more efficient methods can be used instead.

hard to troubleshoot assignment errors

Response:

This is true and we now discuss this limitation in the updated manuscript.

Changed text:

- In *discussion* section: The batch effects between the datasets make it more difficult to troubleshoot errors. Generally, it will be hard to resolve whether mistakes in the constructed tree are caused by the erroneous alignment of datasets or by mismatching created by scHPL.

I think a more modest, focused title should be regarded as a requirement.

We would very much like to keep the title since we believe it accurately describes the proposed method's ability to progressively learn cell identities from a growing set of single cell RNA-seq datasets.